# Rethinking Depth Pruning for Vision Transformers:
# A Heterogeneity-Aware Perspective

**Zhenfeng Su** [1 2]  **Kang Zhao** [3]  **Han Bao** [2]  **Tao Yuan** [2]  **Zhongzhe Hu** [2]  **Xianzhi Yu** [2]  **Wenxuan Wang** [4]

## Abstract

While prior studies have successfully compressed vision Transformers (ViTs) through various pruning techniques, most have concentrated on width pruning to achieve significant reductions in model size. Depth pruning, which removes entire layers from a ViT, is notoriously difficult for accuracy recovery despite its potential to deliver higher speedups, limiting the acceleration achieved by existing joint width-and-depth pruning methods. In this work, we reveal that the failure of existing depth pruning methods lies in their neglect of heterogeneity between different layers, and we introduce HetDPT, a heterogeneity-aware depth pruning method that avoids dimension mismatch. Comprehensive experiments on ImageNet-1K, CIFAR-100, COCO, and ADE20K validate our method: HetDPT achieves a $1.58\times$ speedup for DeiT-B while maintaining accuracy and a $1.39\times$ speedup for DeiT-S with nearly no accuracy degradation. Furthermore, when combined with width pruning, HetDPT+ sets a new state-of-the-art record in extreme ViT pruning, enhancing the acceleration ratio from $4.24\times$ to $5.19\times$ for the Isomorphic-Pruning-2.6G configuration while maintaining near-lossless accuracy; our code is available at https://github.com/Efficient-AI-for-All/HetDPT.

## 1. Introduction

Vision Transformers (ViTs) (Kolesnikov et al., 2021; Liu et al., 2021; Touvron et al., 2021; Han et al., 2021; Chen et al., 2021a; Li et al., 2022; Wang et al., 2022) have demon-

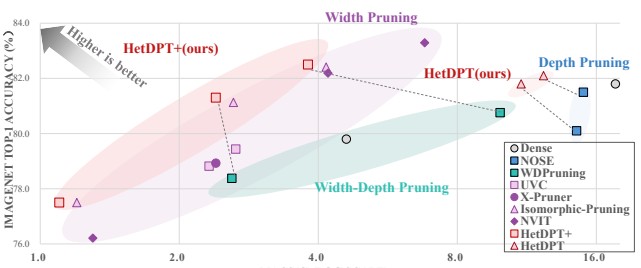

*Figure 1.* Compared with other work, our HetDPT and HetDPT+ offer a state-of-the-art accuracy-speedup Pareto frontier.

strated remarkable performance across various domains. However, their large parameter counts and high computational costs lead to extended inference latency. Structured pruning (He & Xiao, 2023) is effective for model compression and is generally easier to translate into practical speedup on mainstream hardware, while unstructured pruning often requires specialized sparse support, especially to realize benefits at very high sparsity levels.(Pietroń et al., 2023)

**Depth pruning's superiority and limitation.** As a kind of structured pruning, depth pruning denotes removing entire layers from ViTs. Compared to width pruning which only reduces part of channels or attention heads inside a layer, depth pruning delivers significantly higher speedups under equivalent sparsity budgets, as shown in Figure 2 (a). However, depth pruning typically incurs substantial accuracy degradation, especially with aggressive layer removal. Consequently, comprehensive methods that integrate both width and depth pruning are also limited by the depth pruning challenges, resulting in suboptimal performance.

**The missing link: heterogeneity in depth pruning.** While prior research attributes the accuracy collapse in depth pruning to coarse granularity (He & Xiao, 2023; Mao et al., 2017), we challenge this perspective. Our analysis reveals that the true bottleneck arises from the neglect of heterogeneity when pruning layers in a ViT. As shown in Figure 7, individually pruning attention layers or activation function layers leads to drastic accuracy drops at high pruning ratios. Here, an activation function layer refers to the GELU non-linearity inside the FFN, treated as a structural operator con-

---

[1]School of Data Science, The Chinese University of Hong Kong, Shenzhen, China [2]Huawei Technologies Co., Ltd., Shenzhen, China [3]Tsinghua University, Beijing, China [4]School of Information, Renmin University of China, Beijing, China. Correspondence to: Kang Zhao <zhaok14@tsinghua.org.cn>.

*Proceedings of the $43^{rd}$ International Conference on Machine Learning*, Seoul, South Korea. PMLR 306, 2026. Copyright 2026 by the author(s).

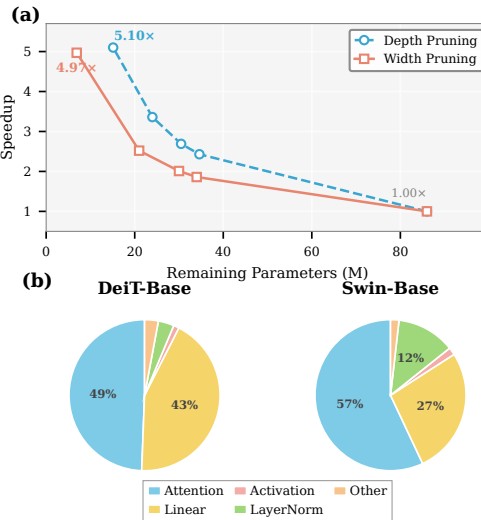

Figure 2. Practical inference speed analysis of ViTs. (a)Speedup comparison: Depth pruning exhibits significantly higher speedup efficiency than width pruning. (b)Latency breakdown of ViTs.

trolled by a binary mask, rather than as a parameterized layer with trainable weights. In contrast, our **Het**erogeneity-aware **DePT**h pruning (HetDPT), which simultaneously prunes distinct layer types while addressing their heterogeneity, significantly improves accuracy retention while maintaining efficiency. A detailed analysis on heterogeneity is provided in Section 2.

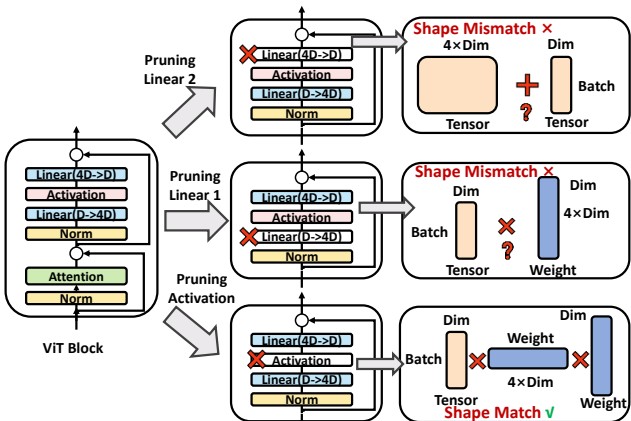

Figure 3. The visualization of dimension mismatch.

**Dimension mismatch bottlenecks.** In heterogeneous depth pruning, the two most time-consuming layer types in ViTs are generally attention layers and linear layers, which together account for over 50% of total inference time, as shown in Figure 2 (b). However, simultaneously pruning attention layers and linear layers creates *dimension mismatch*. As illustrated in Figure 3, if the first linear layer of a feedforward network (FFN) block in ViTs is removed, the output

tensor from the previous attention layer cannot be passed through the second linear layer in the FFN. This is because FFNs employ the expansion-contraction mechanism, e.g., $D \rightarrow 4D \rightarrow D$, where $D$ represents the input channel dimension. Similarly, pruned second linear layers prevent the output tensor from passing through the subsequent attention layers. In a word, dimension mismatch renders heterogeneously depth-pruned ViTs unworkable.

**Contribution.** To address heterogeneity in depth pruning while avoiding dimension mismatch, our contributions are threefold:

- **Pruning activation functions in ViTs.** We tackle the dimension mismatch by proposing strategies to remove activation function layers between linear layers. This allows for the natural merging of adjacent linear layers, reducing model depth while seamlessly aligning dimensions. To the best of our knowledge, we are the **first** to identify and mitigate the redundancy of activation layers in ViTs.

- **The heterogeneity in depth pruning is revealed and addressed.** We provide the first comprehensive heterogeneity analysis including **gradient disparity** and **recovery asymmetry**—two critical phenomena overlooked in prior literature. Guided by these insights, we introduce HetDPT, a two-stage pruning framework equipped with a model accuracy predictor to effectively manage heterogeneity. Notably, we provide **theoretical analysis** to guarantee the effectiveness of our method.

- **Two key state-of-the-art records are established.** With HetDPT, the depth-pruned DeiT-base achieves up to 1.6x speedup while maintaining lossless accuracy, which is the state-of-the-art among depth pruning works. More importantly, building on HetDPT, we further present HetDPT+, a depth-width pruning pipeline that **establishes a new state-of-the-art benchmark for extreme ViT compression**, as demonstrated in Figure 1. HetDPT+ enhances the ViT inference speedup from $4.24\times$ to $5.19\times$ for the Isomorphic-Pruning-2.6G configuration while achieving near-lossless accuracy.

## 2. Insights on Heterogeneity

Pruning redundant activation function layers alongside attention layers is non-trivial. Conventional importance metrics, whether gradient-based or handcrafted, are often insufficient in addressing the heterogeneity between attention layers and activation function layers. Specifically, We highlight two key observations embodying heterogeneity that simultaneously constitute critical challenges in designing an effective heterogeneous depth pruning method.

**Observation 1: gradient disparity.** The attention layers

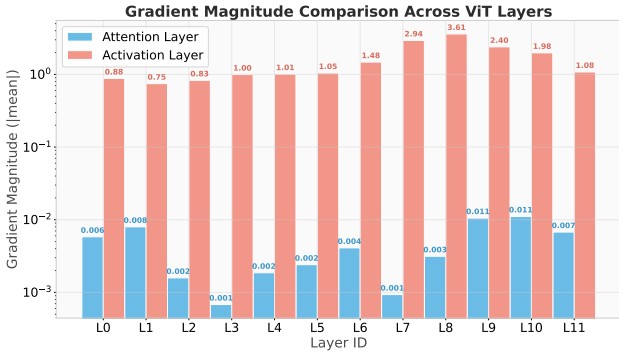

*Figure 4.* Backpropagation gradient magnitudes across attention and activation layers.

**and activation function layers exhibit significant differences in gradient scales during backpropagation, which can lead to biased importance estimations and suboptimal pruning decisions when employing training-based search strategies.** To be specific, we assign each layer in a ViT a learnable importance weight parameter and record the gradient magnitudes for each layer after one epoch of forward and backward propagation. As illustrated in Figure 4, the gradient magnitudes of activation function layers substantially exceed those of attention layers, a trend that persists across increased training epochs.

**Challenge 1: failure of gradient-based pruning metrics.** Given the unique training dynamics in ViTs, using gradients to evaluate the importance of these two types of layers results in significantly biased outcomes, with all attention layers constantly outweighing activation layers.

**Observation 2: recovery asymmetry. Attention layer pruning causes moderate initial accuracy drops but requires extensive retraining for recovery, while activation function layer pruning results in severe initial degradation (often ≥ 90%) but enables rapid recovery during fine-tuning.** To exemplify this phenomenon, we individually prune each layer type in DeiT-S and record accuracy before and after 10-epoch fine-tuning. Figure 5(a) demonstrates that pruning activation function layers leads to catastrophic accuracy declines, with most layers retaining only 0.1% accuracy. Conversely, pruning attention layers results in significantly milder degradation, with most layers experiencing less than 5% accuracy loss. However, as shown in Figure 5(b), after just 10 epochs of fine-tuning, the accuracy of both layer types converges to comparable levels, highlighting the rapid recovery capability of activation function layers following pruning.

Importantly, this single-layer analysis should not be interpreted as evidence that attention-only pruning is optimal. It only characterizes local pruning behavior when all other layers remain intact. In multi-layer pruning, repeatedly remov-

ing attention layers progressively weakens token-mixing capacity across depth, whereas removing activation layers preserves the linear FFN path and mainly simplifies local nonlinearity. Therefore, the optimal strategy under a global pruning budget is generally budget-dependent and requires mixed allocation across layer types, as verified by Figure 7.

**Challenge 2: failure of short-sighted pruning metrics.** The substantial asymmetry in accuracy dynamics between these two types of components in ViTs renders conventional handcrafted pruning metrics ineffective for heterogeneous depth pruning, as these metrics typically reflect only immediate post-pruning accuracy retention.

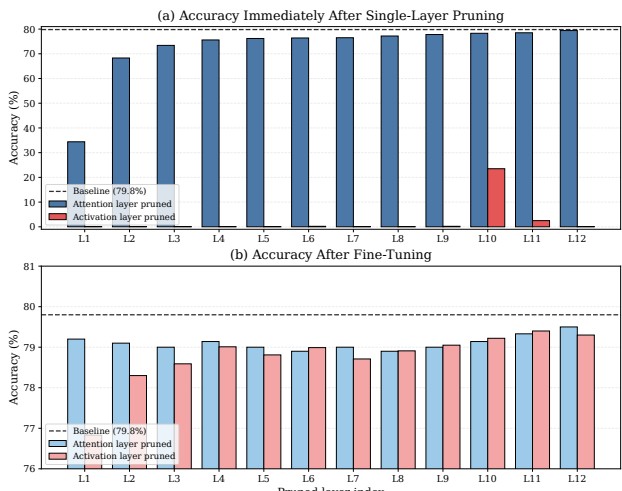

*Figure 5.* Recovery asymmetry in DeiT-Small under single-layer pruning. (a) Top-1 accuracy immediately after pruning an individual attention layer or activation layer at different depths. (b) Top-1 accuracy after 10 epochs of fine-tuning following the pruning of a single attention layer or activation layer at different depths.

## 3. Methods

**Overview of HetDPT.** Our method prunes heterogeneous layers of attention and activation function through a two-stage pipeline: (1) identification of redundant layers, and (2) optimization of the pruned model via fine-tuning and layer merging to accelerate inference. A detailed visualization of the HetDPT pipeline is provided in Figure 6.

**Key design principles from insights on heterogeneity.** Our method is motivated by two principles: 1) Avoid direct cross-type layer importance comparison, especially when using gradient-based metrics. 2) Layer importance should be evaluated based on the final accuracy of fine-tuned pruned ViTs, rather than the immediate accuracy after pruning.

**Alignment of HetDPT with design principles.** The Stage 1 of our method is further divided into two steps: pruning budget allocation (Step 1) and specific redundant layer

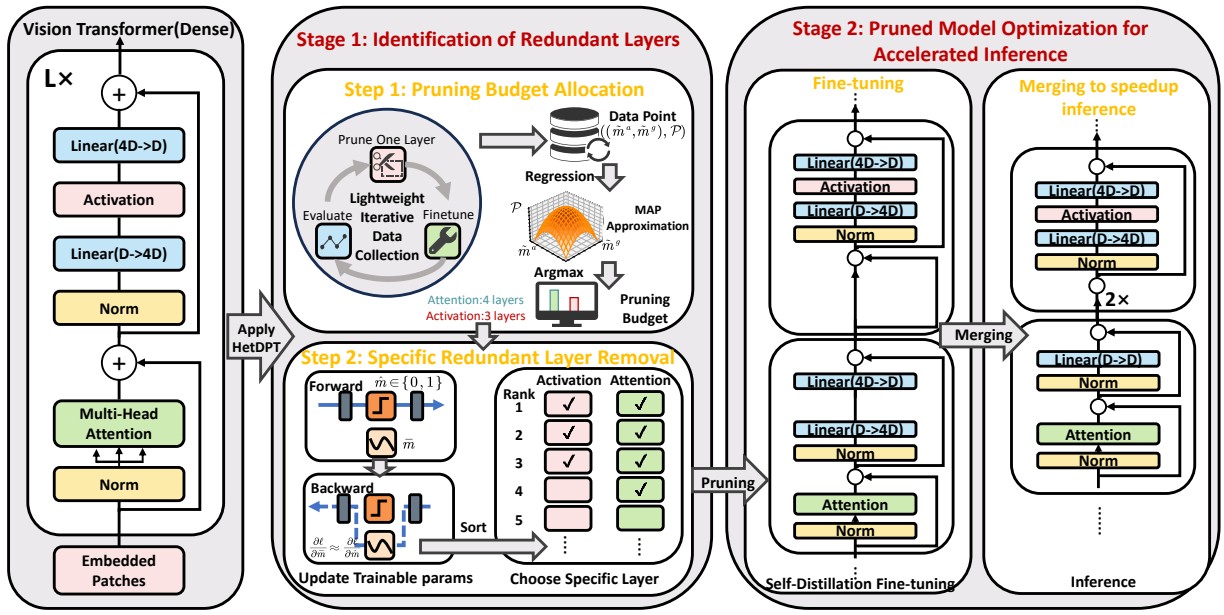

*Figure 6.* Overview of HetDPT: a two-stage depth compression pipeline.

removal (Step 2). 1) Step 1 utilizes a model accuracy predictor, to help establish the optimal quantities of attention and activation function layers to prune based on the accuracy recovered after fine-tuning, directly implementing Principle 2. 2) Importantly, Step 1 focuses solely on determining the pruning quantities for each layer type, avoiding cross-type importance comparisons, thus adhering to Principle 1. 3) Step 2 employs gradient-based methods only within homogeneous layer groups, i.e., either attention or activation function layers, ensuring compliance with Principle 1 throughout the pruning process.

### 3.1. Problem Formulation

Assume a ViT $\mathcal{F}$ has $L$ Transformer blocks. The $l$-th Transformer block $f_l$ typically integrates: 1) two linear layers, i.e., $\mathcal{L}_{l,1}$ and $\mathcal{L}_{l,2}$, 2) an attention layer $\mathcal{A}_l$, 3) a GELU activation function layer $\mathcal{G}_l$. For simplicity, we omit auxiliary components such as residual connections and layer normalization. Thus,

$$f_l(\cdot) := \mathcal{L}_{l,2} \circ \mathcal{G}_l \circ \mathcal{L}_{l,1} \circ \mathcal{A}_l \circ f_{l-1}(\cdot), \quad (1)$$

where $\circ$ denotes the function composition operation. As evident from Figure 2(b), the primary computational overhead in Transformer blocks stems from the attention layers and linear layers. We aim to reduce computational overhead through pruning both attention layers and redundant GELU

activation function layers. That is,

$$\begin{aligned}
f_l^{\hat{m}}(\cdot) &:= \mathcal{L}_{l,2} \circ h\left(\mathcal{G}_l\right) \circ \mathcal{L}_{l,1} \circ h\left(\mathcal{A}_l\right) \circ f_{l-1}^{\hat{m}}(\cdot), \\
h\left(\mathcal{G}_l\right) &= \hat{m}_l^g \circ \mathcal{G}_l + \left(1 - \hat{m}_l^g\right) \circ I, \\
h\left(\mathcal{A}_l\right) &= \hat{m}_l^a \circ \mathcal{A}_l + \left(1 - \hat{m}_l^a\right) \circ I, \\
\hat{m}_l^a, \hat{m}_l^g &\in \{0,1\}. \\
M &:= \left(\{\hat{m}_l^a\}_{l=1}^L, \{\hat{m}_l^g\}_{l=1}^L\right)
\end{aligned} \quad (2)$$

Notably, $\hat{m}_l^g$ and $\hat{m}_l^a$ are the binary masks. A mask's value of 1 preserves the corresponding layer, while 0 prunes it. $I$ denotes the identity mapping, i.e., a direct skip connection when the layer is pruned. $M$ is the set of all the binary masks. A ViT with such masks is referred to as $\mathcal{F}^{\hat{m}}$. Thus, the heterogeneous depth pruning problem is formulated as:

$$\begin{aligned}
\underset{W,M}{\arg\min} \; &\ell\left(\mathcal{F}^{\hat{m}}(X, W, M), Y\right), \\
\text{s.t.} \quad &\tilde{m}^a + \tilde{m}^g = k/L, \\
\tilde{m}^a = \frac{1}{L}\sum_{l=1}^L \mathbf{1}\{\hat{m}_l^a = 0\}, \quad &\tilde{m}^g = \frac{1}{L}\sum_{l=1}^L \mathbf{1}\{\hat{m}_l^g = 0\}.
\end{aligned}$$
$$(3)$$

where $(X, Y)$ are respectively samples and labels of fitting data, and $W$ is the weights of $\mathcal{F}^{\hat{m}}$. $\ell(\cdot)$ refers to the loss function to measure data fitting quality, e.g., cross-entropy loss for classification tasks. $k$ is the sparsity constraint that determines the total number of layers, i.e., the sum of attention layers and activation function layers, to be pruned after depth pruning. $\tilde{m}^a$ and $\tilde{m}^g$ represent the global pruning ratios of attention layers and activation function layers,

respectively. Thus the constrained feasible set $\mathcal{D}_k$:

$$\mathcal{D}_k = \left\{ (\tilde{m}^a, \tilde{m}^g) \in \mathcal{D} : \tilde{m}^a + \tilde{m}^g = k/L \right\}.$$

$\mathcal{D}$ is the admissible ratio set $\mathcal{D} = \{0, \frac{1}{L}, \ldots, 1\}^2$.

## 3.2. Stage 1: Identification of Redundant Layers

**Pruning budget allocation.** Given a sparsity constraint, the primary question is to determine how many attention layers versus activation function layers should be pruned. Naturally, the allocation scheme should maximize the accuracy of depth-pruned Transformers, i.e.,

$$(\tilde{m}^a, \tilde{m}^g) = \underset{\tilde{m}^a, \tilde{m}^g, \Theta}{\arg \max} \, \mathcal{P}(\tilde{m}^a, \tilde{m}^g; \Theta), \qquad (4)$$

where we call $\mathcal{P}(\tilde{m}^a, \tilde{m}^g; \Theta)$ the **Model Accuracy Predictor (MAP)**, which predicts the model's accuracy given $(\tilde{m}^a, \tilde{m}^g)$. $\Theta$ represents the parameters of the MAP itself. **The discreteness of $\tilde{m}^a$ and $\tilde{m}^g$ creates a very finite solution space. Once MAP is properly characterized, we can easily traverse all combinations of $(\tilde{m}^a, \tilde{m}^g)$ to obtain the optimal solution $(\tilde{m}^{a*}, \tilde{m}^{g*})$ maximizing MAP values. Polynomial approximation of MAP.** Generally, it is intractable to obtain an exact expression for the MAP, as determining the precise expression requires a substantial amount of experimental data, and the cost of each training session is high. Here, we approximate the MAP through polynomial fitting combined with rapid data collection. Specifically, we model the MAP function as a polynomial of degree $\kappa$:

$$\mathcal{P}(\tilde{m}^a, \tilde{m}^g; \Theta) = \sum_{i+j \leq \kappa} \theta_{ij}(\tilde{m}^a)^i (\tilde{m}^g)^j, \qquad (5)$$

where $\Theta = \{\theta_{ij}\}$ represents the set of learnable coefficients. By approximating the expression of the MAP using a polynomial, the MAP becomes resolvable via regression (Tyagi et al., 2022).

**Light-weighted data collection for MAP regression.** Although MAP can be approximated using a polynomial, collecting regression data remains time-consuming due to high retraining costs for each pruned ViT configuration. To efficiently collect $((\tilde{m}^a, \tilde{m}^g), \mathcal{P})$ data, we design a lightweight data-collection procedure: 1) We employ an iterative prune $\rightarrow$ fast-finetune $\rightarrow$ evaluate cycle on a representative subset of the training data. Crucially, each subsequent $(\tilde{m}^a, \tilde{m}^g)$ builds incrementally upon the previous one by pruning only a single **additional** layer, which enables direct weight inheritance from the previously fine-tuned model. Such continuity permits rapid accuracy recovery with minimal fine-tuning (typically 10 epochs), avoiding the computational burden of training each pruned ViT from scratch. The resulting dataset, though compact, still maintains high fidelity to the full accuracy landscape. Notably, Transfer entropy (TE) (Lin et al.,

2024) is utilized as the metric to iteratively prune the model. 2) To ensure the collected data are representative, we design two sampling algorithms, including single-type progressive pruning and interleaved pruning. For algorithmic details, see Algorithm 1 and Algorithm 2 in Appendix C. Further details regarding data collection and MAP polynomial approximation, including degree selection and overhead, are provided in Appendix F.

**Specific redundant layer removal.** Having obtained pruning budget allocation scheme, we employ a learning-based mechanism to identify which specific layers should be pruned within each homogeneous layer group. A learnable importance parameter $\bar{m}$ is assigned to each candidate layer. In forward pass, the binary mask $\hat{m}$ in Eq. 2 controls either an activation function layer or an attention layer preservation. In backward pass, since $\hat{m}$ is not trainable, we propagate gradients of $\hat{m}$ to the learnable $\bar{m}$. That is:

$$\frac{\partial \ell}{\partial \bar{m}^a} \approx \frac{\partial \ell}{\partial \hat{m}^a}; \frac{\partial \ell}{\partial \bar{m}^g} \approx \frac{\partial \ell}{\partial \hat{m}^g}. \qquad (6)$$

We then train the model, updating the importance parameters through backpropagation. Layers with the lowest importance scores within their own layer type are progressively pruned until the cumulative pruned layer count matches the target pruning budget. By avoiding cross-type importance score comparisons, this approach **addresses the challenge from gradient disparity.**

## 3.3. Stage 2: Pruned Model Optimization

**Fine-tuning.** After the multiple attention layers and activation function layers with the smallest $\hat{m}$ have been removed, fine-tuning is performed to restore the accuracy. In HetDPT, we can optionally enable self-distillation during fine-tuning, which means conducting knowledge distillation (Hinton et al., 2015) under the guidance of the original ViT to boost the pruned model's accuracy. Note that we only use the original unpruned ViT as the teacher model, without introducing any extra models. This setup aligns with the configuration adopted in related works that also employ single-model methodologies.

**Merging to speed up inference** After fine-tuning, the adjacent linear layers without an activation function layer in between are merged. The resulting Transformer block is thus formulated as:

$$\begin{aligned} f_l^{\hat{m}}(\cdot) &:= \mathcal{L}_{l,2} \circ I \circ \mathcal{L}_{l,1} \circ h\left(\mathcal{A}_l\right) \circ f_{l-1}^{\hat{m}}(\cdot) \\ &= \mathcal{L}_l \circ h\left(\mathcal{A}_{\theta_l}\right) \circ f_{l-1}^{\hat{m}}(\cdot), \end{aligned} \qquad (7)$$

where $\mathcal{L}_l$ is the newly derived linear layer that has the same number of input channels as $\mathcal{L}_{l,1}$ and the same number of output channels as $\mathcal{L}_{l,2}$. In this manner, **the depth of a ViT is further reduced with rigorous dimension alignment among layers.** Besides, following such merging, the pruned

ViT achieves significant inference speedups while its accuracy remains exactly the same as it was prior to merging.

### 3.4. Theoretical Analysis of MAP

We provide a theoretical justification for the effectiveness of MAP, summarized in three aspects (detailed proofs are given in Appendix F.5).

**Existence of Optimum.** Since the feasible set $\mathcal{D}_k$ is finite and the accuracy metric is bounded, a global optimizer $(\tilde{m}^{a\star}, \tilde{m}^{g\star})$ is guaranteed to exist, ensuring the search problem is well-posed.

**Theorem 3.1** (Existence of an optimal pruning configuration). *For any pruning budget $k$,*

$$(\tilde{m}^{a\star}, \tilde{m}^{g\star}) \in \underset{(\tilde{m}^a, \tilde{m}^g) \in \mathcal{D}_k}{\arg\max} \mathcal{P}(\tilde{m}^a, \tilde{m}^g)$$

*exists.*

**Polynomial Approximation.** We first introduce the following assumption to enable continuous analysis.

**Assumption 3.2** (Continuous Relaxation of Pruning Ratios). Although structured pruning masks $(\hat{m}^a, \hat{m}^g)$ are discrete, their normalized pruning ratios

$$(\tilde{m}^a, \tilde{m}^g) \in \mathcal{D} = \{0, \tfrac{1}{L}, \ldots, 1\}^2 \subset [0,1]^2$$

admit a continuous relaxation in which $(\tilde{m}^a, \tilde{m}^g)$ is treated as a point in the full square $[0,1]^2$. Furthermore, the accuracy $\mathcal{P}$ evaluated on $\mathcal{D}$ extends to a continuous map

$$\tilde{\mathcal{P}} : [0,1]^2 \to [0,1], \qquad \tilde{\mathcal{P}}|_{\mathcal{D}} = \mathcal{P}.$$

By the Stone–Weierstrass theorem(De Branges, 1959), any continuous function on a compact domain can be uniformly approximated by polynomials. This justifies our use of a polynomial-based MAP to capture the accuracy landscape $\mathcal{P}(\tilde{m}^a, \tilde{m}^g)$.

**Theorem 3.3** (Uniform Polynomial Approximation of the Accuracy Surface). *Under Assumption 3.2, for every $\varepsilon > 0$ there exists a polynomial $Q(x, y) \in \mathbb{R}[x, y]$ such that*

$$\sup_{(x,y) \in [0,1]^2} \left| \tilde{\mathcal{P}}(x,y) - Q(x,y) \right| < \varepsilon.$$

*Consequently, the approximation error on the discrete domain is also bounded*

$$\max_{(\tilde{m}^a, \tilde{m}^g) \in \mathcal{D}} \left| \mathcal{P}(\tilde{m}^a, \tilde{m}^g) - Q(\tilde{m}^a, \tilde{m}^g) \right| < \varepsilon.$$

**Robustness to Fast Finetuning.** In practice, MAP is trained on accuracy observations obtained via fast finetuning, which introduces a systematic bias $b$ and random noise $\varepsilon$. The following lemma shows that constant bias does not alter the optimal configuration.

**Lemma 3.4** (Bias preserves the maximizer). *For any objective function $\mathcal{P}(x)$ and a constant $b$, the set of maximizers remains unchanged:*

$$\arg\max_x \mathcal{P}(x) = \arg\max_x \left( \mathcal{P}(x) + b \right).$$

Consequently, MAP trained on fast-finetuned data consistently identifies the optimal pruning configuration of the ground-truth accuracy surface.

**Theorem 3.5** (Consistency of MAP). *Let $y = \mathcal{P}(x) + b + \varepsilon$ be noisy observations of the true accuracy $\mathcal{P}(x)$, where $b$ is a constant bias and $\varepsilon$ is zero-mean noise independent of $x$. If MAP is trained via least squares, then: (i) the noise variance does not affect gradient-based optimization; (ii) the learned predictor $f^*$ satisfies $\arg\max_x f^*(x) = \arg\max_x \mathcal{P}(x)$ for the lemma 3.4.*

**Summary.** These results collectively establish the theoretical soundness of MAP: it searches for a guaranteed optimum (Theorem 3.1) using a provably expressive polynomial approximator (Theorem 3.3) that remains statistically consistent even when trained on noisy, low-cost signals (Theorem 3.5).

### 3.5. HetDPT+

Upon completion of the two-stage pipeline, we obtain a depth-pruned ViT that demonstrates enhanced computational efficiency while maintaining competitive accuracy. To further explore the potential of HetDPT in extreme ViT compression scenarios, we integrate the depth-pruned model with established width pruning methodologies (Fang et al., 2024), thereby proposing HetDPT+. HetDPT+ represents a comprehensive pruning framework that addresses both depth and width dimensions of ViT architectures. The detailed performance is presented in Experiments.

## 4. Experiments

We evaluate our method on four well-established benchmarks: (1) CIFAR-100 (Krizhevsky et al., 2009), containing 50,000 training and 10,000 test images across 100 categories; (2) ImageNet-1k (Deng et al., 2009), a large-scale classification dataset with 1.28 million training and 50,000 validation samples spanning 1,000 classes; (3) COCO2017 (Lin et al., 2014), a widely-used object detection benchmark with 118,000 training and 5,000 validation images annotated with 80 object categories; (4) ADE20K (Zhou et al., 2017), a semantic segmentation benchmark comprising 20,000 training and 2,000 validation images with 150 semantic categories. Inference profiling (MACs/throughput) is conducted on an H800 PCIE GPU. More experimental details are provided in our Appendix A.

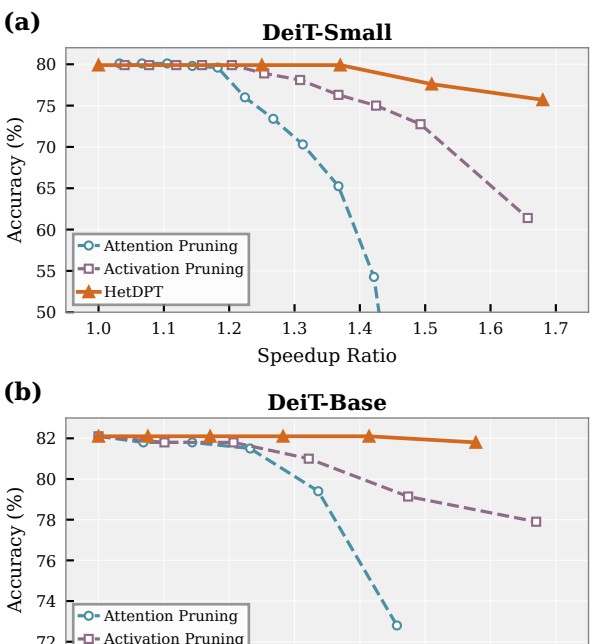

**(a)**

**(b)**

*Figure 7.* Comparison of HetDPT and single-layer-type pruning on ImageNet-1K for DeiT-S and DeiT-B. (a) Top-1 accuracy vs. speedup on DeiT-S. (b) Top-1 accuracy vs. speedup on DeiT-B.

*Table 1.* Comparison with depth pruning, depth-width (DW) pruning on ImageNet-1k.

| Method | Top-1 (%) | MACs (G) | Throughput (img/s) | Params (M) | Speedup (×) |
|---|---|---|---|---|---|
| DeiT-B (Dense) | 81.8 | 17.6 | 741.5 | 86.6 | 1.00 |
| DW Prun. (0.2-12) | 81.9 | 13.6 | 907.6 | 68.2 | 1.22 |
| DW Prun. (0.2-10) | 80.9 | 10.8 | 1037.9 | 59.4 | 1.40 |
| NOSE (5 layers) | 81.8 | 14.6 | 894.7 | 75.5 | 1.21 |
| NOSE (6 layers) | 81.5 | 14.1 | 921.4 | 73.2 | 1.24 |
| **HetDPT (8 layers)** | **82.1** | 11.8 | 1048.5 | 61.4 | 1.41 |
| **HetDPT (10 layers)** | 81.8 | **10.5** | **1169.7** | **54.9** | **1.58** |
| DeiT-S (Dense) | 79.8 | 4.6 | 2349.9 | 22.1 | 1.00 |
| DW Prun. (0.3-12) | 78.6 | 3.1 | 2989.8 | **15.0** | 1.27 |
| NOSE (5 layers) | 79.6 | 3.7 | 2875.3 | 19.5 | 1.22 |
| **HetDPT (6 layers)** | **80.1** | 3.3 | 2987.3 | 17.6 | 1.27 |
| **HetDPT (8 layers)** | 79.7 | **3.0** | **3275.0** | 15.9 | **1.39** |
| Swin-S (Dense) | 83.2 | 8.7 | 1106.9 | 49.6 | 1.00 |
| DW Prun. (0.2-24) | 82.2 | 6.8 | 1140.1 | 39.2 | 1.10 |
| DW Prun. (0.2-22) | 81.8 | 6.3 | 1273.0 | 34.4 | 1.15 |
| **HetDPT (10 layers)** | **82.4** | 7.3 | **1365.9** | 43.9 | **1.23** |
| Swin-B (Dense) | 83.5 | 15.4 | 690.25 | 87.8 | 1.00 |
| SAViT | 82.6 | 7.7 | 1056.08 | 33.0 | 1.53 |
| **HetDPT** | **82.9** | 9.8 | **1106.95** | 55.3 | **1.60** |

drop. On Swin Transformers, despite their more parameter-efficient hierarchical design, HetDPT still delivers consistent gains, achieving 1.23× speedup on Swin-S and 1.6× on Swin-B with minimal accuracy degradation.

*Table 2.* Comparison with SOTA ViT compression methods on ImageNet-1k.

| Method | Top-1 (%) | MACs (G) | Throughput (img/s) | Params (M) | Speedup (×) |
|---|---|---|---|---|---|
| DeiT-B-KD | 83.3 | 17.7 | 741.5 | 87.3 | 1.00 |
| NViT-S | 82.2 | 3.9 | 2394.5 | 20.8 | 3.23 |
| Iso-Prun. 4.2G | 82.4 | 4.2 | 2516.0 | 20.7 | 3.39 |
| **HetDPT+ 3.7G** | **82.5** | **3.7** | **2763.9** | **20.0** | **3.73** |
| Iso-Prun. 2.6G | 81.1 | 2.6 | 3145.8 | 13.1 | 4.24 |
| **HetDPT+ 2.4G** | **81.4** | **2.4** | **3845.6** | 13.2 | **5.19** |
| NViT-T | 76.2 | 1.2 | 6013.7 | 6.7 | 8.11 |
| Iso-Prun. 1.2G | 77.5 | 1.2 | 6387.4 | **5.7** | 8.61 |
| **HetDPT+ 1.1G** | 77.4 | **1.1** | **6814.0** | 6.2 | **9.19** |

## 4.1. Main Results

**Accuracy-speedup tradeoff analysis.** We compare HetDPT against isolated pruning of attention layers and activation layers using NOSE (Lin et al., 2024). As shown in Figure 7, HetDPT consistently achieves superior accuracy at equivalent speedup ratios. For DeiT-S, HetDPT attains 1.39× speedup with 79.6% accuracy, substantially outperforming attention-only pruning (70.3%) and activation-only pruning (76.3%). For DeiT-B, HetDPT achieves 1.58× speedup while fully preserving baseline accuracy (81.8%), which highlights the benefits of considering heterogeneity in our method.

**Comparison with depth pruning and depth-width pruning on ImageNet-1k.** As shown in Table 1, HetDPT consistently outperforms existing depth pruning and depth-width (DW) pruning methods across all architectures. On DeiT-B, HetDPT achieves 82.1% Top-1 accuracy with only 11.8G MACs, surpassing the baseline (81.8%@17.6G) while reducing computation by 33%. With 10 layers pruned, HetDPT matches the baseline accuracy while achieving 1.58× speedup—substantially outperforming NOSE, which prunes fewer layers yet yields inferior efficiency. Similar improvements are observed on DeiT-S, where HetDPT with 8 layers pruned achieves 1.39× speedup with only 0.1% accuracy

**Comparison with SOTA ViT compression methods on ImageNet-1k.** As shown in Table 2, we compare HetDPT+ against state-of-the-art extreme compression methods. At comparable accuracy levels, HetDPT+ consistently achieves higher speedup than Isomorphic Pruning: 3.73× vs. 3.39× at ∼82.5% accuracy, and 5.19× vs. 4.24× at ∼81% accuracy. For ultra-extreme compression, HetDPT+ achieves 77.4% accuracy with only 1.1G MACs and 9.19× speedup, demonstrating that our depth pruning approach enables existing compression techniques to reach new state-of-the-art

performance.

**Transfer learning on CIFAR.** As illustrated in Tab. 3, Het-DPT exhibits transfer learning performance equivalent to or superior to related works. This indicates that the architecture-induced invariance learned by HetDPT on ImageNet remains robust to domain shifts, and does not suffer from catastrophic forgetting during fine-tuning.

*Table 3.* Transfer learning results (CIFAR-100).

| Method | Fine-tuning (%) | Linear probing (%) |
|---|---|---|
| DeiT-B (Dense) | 90.5 | 80.6 |
| Evo-ViT | 90.1 | 79.1 |
| EViT | 90.0 | 80.2 |
| TPS | 90.1 | 76.5 |
| NOSE (5 layers) | 90.3 | 81.3 |
| NOSE (6 layers) | 90.2 | 80.6 |
| **Ours (8 layers)** | **90.4** | **81.2** |
| **Ours (10 layers)** | 90.2 | 80.6 |

*Table 4.* Results on COCO2017 (12-epoch training).

| Backbone | mAP (%) | AP50 (%) | AP75 (%) |
|---|---|---|---|
| DeiT-B (Dense) | 32.8 | 50.0 | 34.4 |
| NOSE (6 layers) | 32.0 | 48.7 | 33.6 |
| **Ours (10 layers)** | **32.4** | **49.6** | **33.9** |

*Table 5.* Results on ADE20k.

| Method | mIoU (%) | mAcc (%) | aAcc (%) |
|---|---|---|---|
| DeiT-B (Dense) | 47.0 | 57.5 | 82.6 |
| EViT | 45.5 | 55.9 | 81.9 |
| TPS | 45.3 | 55.1 | 81.9 |
| NOSE (5 layers) | 46.2 | 56.5 | 82.2 |
| NOSE (6 layers) | 45.6 | 55.2 | 82.0 |
| **Ours (8 layers)** | **46.4** | **56.5** | **82.2** |
| **Ours (10 layers)** | 45.4 | 55.7 | 81.9 |

**Result on COCO object detection.** To further validate the transferability of HetDPT on dense prediction tasks, we evaluate our method on COCO2017 object detection using the DETR framework. As shown in Table 4, HetDPT with 10 layers pruned achieves 32.4% mAP, with only a 0.4% drop from the DeiT-B baseline performance. Compared to NOSE with only 6 layers pruned, HetDPT removes more layers yet still achieves superior detection performance.

**Result on ADE20k.** We also extend the proposed method to the dense prediction task on ADE20k. As shown in Table 5, when pre-trained on ImageNet-1k, our model with 8 layers removed exhibits a minimal gap of only approximately 0.6% compared to the baseline. Also, HetDPT consistently outperforms other approaches.

*Table 6.* Results of DINO-V2-Giant on CIFAR-100

| Method | Top-1 (%) | Throughput (img/s) | Params (M) | Speedup (×) |
|---|---|---|---|---|
| DINO-V2-Giant (Dense) | 94.91 | 45.16 | 1134.92 | 1.00 |
| NOSE (12 layers) | 94.53 | 53.03 | 1021.55 | 1.17 |
| **HetDPT (15 layers)** | **94.85** | **56.87** | **927.12** | **1.26** |

*Table 7.* Ablation analysis of Stage 1 components.

| Models | TE | Stage1-Step1 | Stage1-Step2 | Acc |
|---|---|---|---|---|
| DeiT-S | ✔ | ✔ | ✔ | **79.7** |
| | ✔ | ✔ | | 79.4 |
| | | | ✔ | 78.5 |
| | ✔ | | | 78.5 |
| DeiT-B | ✔ | ✔ | ✔ | **81.8** |
| | ✔ | ✔ | | 80.5 |
| | | | ✔ | 78.1 |
| | ✔ | | | 78.9 |

**Scalability to DINO-V2-Giant.** To validate the scalability of our method to billion-parameter models, we conduct experiments with DINO-V2-Giant on CIFAR-100, as shown in Table 6. HetDPT prunes 15 layers and attains 94.85% Top-1 accuracy, with only a 0.06% drop from the baseline. Despite removing more layers than NOSE (15 vs. 12), Het-DPT achieves superior accuracy alongside greater parameter reduction and a 1.26× speedup, confirming the scalability of our approach to billion-parameter models. In terms of per-image latency, DINOv2-Giant requires 22.1 ms/image, NOSE requires 18.9 ms/image, and HetDPT requires 17.6 ms/image under the same profiling setting.

**Excellent orthogonality to token pruning methods** Het-DPT exhibits remarkable compatibility with token pruning, as shown in Figure 8 in Appendix B. When combined with GTP-ViT ($num_{prop} = 20$), HetDPT further achieves a 1.24× speedup compared to HetDPT alone, with only a 0.33% accuracy drop (81.47% vs. 81.8%). Meanwhile, we observe that under equivalent accuracy drops, the speedup compared to dense model remain consistently stable, maintaining between 1.5-1.6×, and are even slightly higher compared to methods without token pruning. This demonstrates that HetDPT does not introduce the performance degradation typically associated with token pruning methods.

### 4.2. Ablation Study

We evaluate the contributions of Stage 1's components on DeiT-S (8 layers pruned) and DeiT-B (10 layers pruned). For clarity, we designate pruning budget allocation as "Stage1-Step1", and specific redundant removal as "Stage1-Step2".

**Ablation results.** As shown in Table 7, removing Stage1 Step2 causes significant performance drops: DeiT-B from 81.8% to 80.5% and DeiT-S from 79.7% to 79.4%. Using only TE (directly using TE metric for layer selection)

leads to severe degradation: DeiT-B to 78.9% and DeiT-S to 78.5%. Using only Stage1-Step 2 (directly using trainable importance scores for layer selection) performs even worse: DeiT-B to 78.1% and DeiT-S to 78.5%.

These results confirm that all components are essential: TE and Stage1-Step 1 tackle the challenge incurred by recovery asymmetry through separate layer-type statistics. Together, Stage1-Step1 and Stage1-Step2 mitigate the challenge incurred by gradient disparity via eliminating cross-type comparisons, and account for modeling layer interdependencies.

## 5. Related Work

**Structured ViT pruning** Depending on the pruning targets, existing structured pruning methods for ViTs can be divided into four categories: 1) *width pruning*, such as NViT (Yang et al., 2023), IsomorphicPruning (Fang et al., 2024), and related works (Chavan et al., 2022; Zhu et al., 2021; Yu et al., 2022b; Zheng et al., 2022); 2) *depth pruning*, such as NOSE (Lin et al., 2024); 3) *joint width-depth pruning*, such as WD-Pruning(DW Prun.) (Yu et al., 2022a); and 4) *token pruning/merging*, such as ToMe (Bolya et al., 2023), GTP-ViT (Xu et al., 2024b), DynamicViT (Rao et al., 2021), Evo-ViT (Xu et al., 2022), EViT (Liang et al., 2022), TPS (Wei et al., 2023), and related works (Chen et al., 2023a; Wu et al., 2024). Among the first three categories, width pruning often yields limited speedup, depth pruning provides higher acceleration but is harder to recover, and joint width-depth pruning must balance both trade-offs.

**Other pruning techniques for ViTs.** Besides structured pruning, other Transformer compression techniques include unstructured pruning (Chen et al., 2021b; Frantar & Alistarh, 2023; Sun et al., 2024) and semi-structured pruning (Zhao et al., 2024; Xu et al., 2024a; Lu et al., 2023; Liu et al., 2025). However, these methods usually require specialized hardware support. Another related line is activation pruning (Chen et al., 2023b; Ganguli & Chong, 2024), which dynamically removes neurons or feature maps during inference. In contrast, our work statically removes activation-function layers rather than dynamically pruning activations.

**Related studies beyond ViTs.** Several works beyond ViTs are related to our motivation. DepthShrinker (Fu et al., 2022) studies activation removal and linear-layer merging in CNNs, while recent LLM works (He et al., 2024; Gromov et al., 2024; Men et al., 2025) reveal heterogeneous or layer-wise redundancy mainly through analysis or coarse-grained layer dropping. In contrast, we study finer-grained structured depth pruning for ViTs, jointly modeling attention and activation layers under a unified budget allocation strategy, and analyze their distinct pruning and recovery behaviors.

**Difference from DepthShrinker.** DepthShrinker also removes nonlinear activation functions and merges adjacent linear/convolutional operators, but it is designed for CNN compression where the main pruning candidates are homogeneous feedforward operators. Directly transferring this idea to ViTs does not address the heterogeneous interaction between attention layers and FFN activation layers. In contrast, HetDPT formulates ViT depth pruning as a mixed heterogeneous allocation problem, where attention and activation layers have different gradient scales and recovery dynamics. Our MAP-based budget allocation and within-type layer selection are specifically designed for this setting. Empirically, directly applying a DepthShrinker-style strategy to ViTs leads to substantially lower accuracy than HetDPT.

## 6. Conclusion

We propose HetDPT, a two-stage framework for ViT depth pruning. It addresses gradient disparity and recovery asymmetry via MAP-based budget allocation and separate importance estimation for attention and activation layers. Experiments show that HetDPT achieves state-of-the-art depth pruning performance, while HetDPT+ further improves results in extreme compression settings. A limitation is that MAP is refitted for each architecture family in our current pipeline, and extending HetDPT to language models or other modalities remains future work.

## Acknowledgements

This work was supported in part by Huawei Technologies Co., Ltd. We thank the anonymous reviewers for their constructive comments and suggestions, which helped improve the quality and clarity of this paper.

We also thank the members of the research teams at the School of Data Science, The Chinese University of Hong Kong, Shenzhen, Tsinghua University, and Renmin University of China for valuable discussions and feedback during the development of this work.

The experiments were conducted using computing resources provided by Huawei Technologies Co., Ltd.

## Impact Statement

This work presents HetDPT, a heterogeneity-aware depth pruning method for Vision Transformers that achieves significant computational efficiency improvements while maintaining model accuracy. We discuss the potential broader impacts of this research below.

**Positive societal impacts.** The primary contribution of this work is enabling more efficient deployment of Vision Transformers, which carries several beneficial implications. First, by substantially reducing computational requirements (up

to 9.19× speedup), our method can democratize access to state-of-the-art vision models for researchers and practitioners with limited computational resources. Second, the reduced energy consumption associated with compressed models contributes to more environmentally sustainable AI development—a critical consideration given the growing carbon footprint of large-scale deep learning. Third, efficient ViT models enable real-time deployment on edge devices, facilitating applications in healthcare diagnostics, accessibility tools, and autonomous systems where low latency is essential.

**Potential risks and limitations.** As with any model compression technique, there exists a risk that pruned models may exhibit subtle performance degradation on underrepresented data distributions not captured during evaluation. While our extensive experiments across multiple benchmarks (ImageNet-1k, CIFAR-100, COCO, ADE20K) demonstrate robust generalization, practitioners should conduct thorough validation when deploying compressed models in safety-critical applications. Additionally, the efficiency gains from our method could lower barriers to deploying vision models in surveillance or other privacy-sensitive contexts, which warrants careful consideration of deployment guidelines and regulatory frameworks.

**Final thoughts.** We believe the benefits of efficient vision model deployment—including reduced computational costs, broader accessibility, and environmental sustainability—outweigh the potential risks, particularly when accompanied by responsible deployment practices. Our work does not introduce new capabilities for harmful applications beyond what existing ViT models already enable; rather, it makes existing capabilities more accessible and sustainable.

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

# A. Experiment Setup

**Datasets and Benchmarks.** We evaluate our method on four established benchmarks: (1) CIFAR-100 (Krizhevsky et al., 2009), containing 50,000 training and 10,000 validation images across 100 categories; (2) ImageNet-1k (Deng et al., 2009), a large-scale classification dataset with 1.28 million training and 50,000 validation samples spanning 1,000 classes; (3) COCO2017 (Lin et al., 2014), a widely-used object detection benchmark with 118,000 training and 5,000 validation images annotated with 80 object categories; (4) ADE20K (Zhou et al., 2017), a semantic segmentation benchmark comprising 20,000 training and 2,000 validation images with 150 semantic categories.

**Evaluation Protocol.** Our evaluation framework consists of four components: Primary validation on ImageNet-1k for classification accuracy, cross-task verification on COCO2017 for object detection, cross-task verification using ADE20K for dense prediction capability, and transfer learning analysis on CIFAR-100 to assess feature transferability. All experiments follow standardized evaluation metrics for their respective tasks.

**Training Configuration.** ImageNet-1k training employs 8 NVIDIA A100 GPUs with 128 samples per device, using AdamW optimizer ($\beta_1 = 0.9$, $\beta_2 = 0.999$) and a cosine-decayed learning rate from $5 \times 10^{-5}$ to $1 \times 10^{-5}$. CIFAR-100 experiments utilize SGD optimizer with fixed learning rate 0.1 and 384 batch size per GPU (224×224 resolution). For COCO2017 detection, we use H-Deformable DETR with hybrid branch, training 12 epochs on 8 A100 GPUs (batch size 1/GPU). We employ AdamW optimizer with weight decay 0.05, learning rate $1 \times 10^{-4}$ ($5 \times 10^{-5}$ for backbone). The model uses two-stage detection with 300 one-to-one and 1500 one-to-many queries ($k = 6$, $\lambda = 1.0$), along with box refinement, mixed selection, and look-forward-twice strategy. ADE20K segmentation adopts polynomial learning rate decay (power=1.0) from initial 0.01 using SGD across 8 GPUs (2 images/device).

**Evaluation Details.** To ensure reproducibility and fair comparison, all baseline models are instantiated using the official implementation scripts and pre-trained weights provided by their respective authors. Computational complexity analysis, including FLOPs and parameter count estimation, is conducted using the standardized measurement APIs from the Torch Pruning library(Fang et al., 2024). Inference throughput evaluation follows a systematic protocol: each model undergoes 10 warm-up iterations to stabilize GPU memory allocation and kernel optimization, followed by 100 consecutive inference runs. Throughput is computed as the average inference time over the latter 100 iterations to mitigate the impact of initialization overhead and ensure statistical reliability. All experiments maintain uniform input specifications with RGB images of resolution 224×224 and a consistent batch size of 256 across all evaluated architectures.

**Hardware Platform and Training Time** All training phases are performed on NVIDIA A100 Tensor Core GPUs , while inference profiling (MACs/throughput) is conducted on an NVIDIA H800 Tensor Core GPU (PCIe) unless otherwise specified. Overall, for Deit-Base, the pipeline requires approximately 1.5 hours for pruning–ratio determination, 0.5 hours for layer identification on GPUs, and 28 hours for final fine-tuning on 8 GPUs, amounting to roughly 30 hours of total computation.

# B. More Experiments

### B.1. Main Results

**Accuracy-Speedup Tradeoff Analysis.** We conducted a comprehensive evaluation of HetDPT's heterogeneous pruning efficacy using DeiT-S and DeiT-B architectures. To establish baseline comparisons, we implemented isolated pruning of self-attention layers and activation layers respectively using the NOSE algorithm, followed by 400-epoch retraining for accuracy recovery (consistent with the NOSE setting). Subsequent application of HetDPT's unified pruning framework under identical experimental conditions revealed statistically significant improvements: as shown in Figure 7, HetDPT achieves higher model accuracy at equivalent speedup ratios compared to single-layer-type pruning baselines.

For DeiT-S, HetDPT enables a 1.39× speedup ratio with near-lossless accuracy retention (79.6% vs. baseline 79.9%), outperforming both activation-layer-only pruning (76.3%) and self-attention-only pruning (70.3%). For DeiT-B, HetDPT achieves an enhanced 1.58× acceleration while maintaining identical baseline accuracy (81.8% vs. baseline 81.8%), significantly surpassing activation-layer pruning (77.9%) and self-attention pruning (72.8%).

These empirical results substantiate key advantage of our heterogeneous optimization approach: through synergistic layer interaction, our method achieves superior speedup ratios compared to individual layer-type pruning strategies, demonstrating enhanced capability in eliminating architectural redundancies within Vision Transformers.

*Table 8.* Comparison with depth pruning, depth-width pruning, and token pruning methods on ImageNet-1k.

| Method | Top-1 (%) | MACs (G) | Throughput (images/s) | Params (M) | Speedup (×) |
|---|---|---|---|---|---|
| DeiT-B (Dense) | 81.8 | 17.6 | 741.5 | 86.6 | 1.00 |
| DynamicViT | 81.3 | 11.2 | 1151.5 | 89.5 | 1.55 |
| Evo-ViT | 81.3 | 11.3 | 1147.0 | 87.3 | 1.55 |
| EViT | 81.3 | 11.2 | 1138.1 | 86.6 | 1.53 |
| TPS | 81.4 | 12.9 | 993.9 | 89.5 | 1.34 |
| ToMe | 80.6 | 13.2 | 949.2 | 86.6 | 1.28 |
| GTP-ViT | 81.5 | 13.2 | 955.3 | 86.6 | 1.29 |
| WD-Pruning (0.2-12) | 81.9 | 13.6 | 907.6 | 68.2 | 1.22 |
| WD-Pruning (0.2-10) | 80.9 | 10.8 | 1037.9 | 59.4 | 1.40 |
| NOSE (5 layers) | 81.8 | 14.6 | 894.7 | 75.5 | 1.21 |
| NOSE (6 layers) | 81.5 | 14.1 | 921.4 | 73.2 | 1.24 |
| **HetDPT (8 layers)** | **82.1** | 11.8 | 1048.5 | 61.4 | 1.41 |
| **HetDPT (10 layers)** | 81.8 | **10.5** | **1169.7** | **54.9** | **1.58** |
| DeiT-S (Dense) | 79.8 | 4.6 | 2349.9 | 22.1 | 1.00 |
| TPS | 79.7 | 3.3 | 3051.2 | 22.8 | 1.30 |
| ToMe | 79.5 | 3.3 | 2874.5 | 22.1 | 1.22 |
| GTP-ViT | 79.5 | 3.5 | 2793.4 | 22.1 | 1.19 |
| WD-Pruning (0.3-12) | 78.6 | 3.1 | 2989.8 | **15.0** | 1.27 |
| NOSE (4 layers) | 79.8 | 3.8 | 2774.8 | 20.1 | 1.18 |
| NOSE (5 layers) | 79.6 | 3.7 | 2875.3 | 19.5 | 1.22 |
| **HetDPT (6 layers)** | **80.1** | 3.3 | 2987.3 | 17.6 | 1.27 |
| **HetDPT (8 layers)** | 79.7 | **3.0** | **3275.0** | 15.9 | **1.39** |
| SwinTransformer-S (Dense) | 83.2 | 8.7 | 1106.93 | 49.6 | 1.00 |
| WD-Pruning (0.2-24) | 82.2 | 6.8 | 1140.14 | 39.2 | 1.10 |
| WD-Pruning (0.2-22) | 81.8 | 6.3 | 1272.97 | 34.4 | 1.15 |
| **HetDPT (10 layers)** | **82.4** | 7.3 | **1365.89** | 43.9 | **1.23** |
| SwinTransformer-B (Dense) | 83.5 | 15.4 | 690.25 | 87.8 | 1.00 |
| SAViT | 82.6 | 7.7 | 1056.08 | 33.0 | 1.53 |
| **HetDPT** | **82.9** | 9.8 | **1106.95** | 55.3 | **1.60** |

**Comparison HetDPT with depth pruning, depth-width pruning, and token pruning methods on ImageNet-1k.** To demonstrate the effectiveness of our HetDPT method, we conduct extensive comparisons with existing pruning strategies, including depth pruning, depth-width pruning, and token pruning methods. Since works focusing solely on depth pruning are limited, token pruning baselines are also incorporated for a more comprehensive evaluation. As shown in Table 8, HetDPT consistently outperforms existing methods across multiple backbones.

For DeiT-B, HetDPT achieves 82.1% Top-1 accuracy with 11.8G MACs, surpassing the dense baseline accuracy (81.8%@17.6G) while reducing computation by 30.1%. Notably, when pruning 10 layers, our method maintains baseline accuracy (81.8%) but dramatically improves efficiency, with only 54.9M parameters, 10.5G MACs, and 1169.7 images/s throughput, representing a 1.58× speedup. These results reveal that HetDPT is able to remove redundant computations while preserving essential semantic representations, a clear indication of effective structural pruning.

For DeiT-S, HetDPT achieves nearly lossless compression, reaching 80.1% accuracy with 6 layers or maintaining 79.7% accuracy with 8 layers pruned. In the latter case, it achieves superior parameter efficiency (15.9M vs. 22.1M), reduced computational complexity (3.0G vs. 4.6G), and higher throughput (3275.0 vs. 2349.9 images/s), substantially outperforming competitive token-pruning methods.

Moreover, the results generalize well to other backbones. For Swin-S and Swin-B, HetDPT delivers consistent improvements over WD-Pruning baselines, achieving up to 1.23× speedup on Swin-S. These consistent gains across both plain and other ViTs confirm the versatility and robustness of our method.

**Comparison HetDPT+ with SOTA ViT compression methods and other off-the-shelf ViT models on ImageNet-1k.**

*Table 9.* Comparison with SOTA ViT compression methods and other off-the-shelf ViT models on ImageNet-1k.

| Method | Top-1 (%) | MACs (G) | Throughput (images/s) | Params (M) | Speedup (×) |
|---|---|---|---|---|---|
| DeiT-B-KD | 83.3 | 17.7 | 741.5 | 87.3 | 1.00 |
| Cait-S24 | 83.5 | 8.6 | 800.8 | 46.9 | 1.08 |
| HetDPT–KD | 83.3 | 10.5 | 1169.7 | 54.9 | 1.58 |
| DeiT-S-KD | 81.2 | 4.6 | 2370.6 | 22.1 | 3.20 |
| Swin-T | 81.2 | 4.5 | 1913.24 | 28.3 | 2.58 |
| NViT-S | 82.2 | 3.9 | 2394.5 | 20.8 | 3.23 |
| TNT-S | 81.5 | 4.8 | 1168.9 | 23.8 | 1.58 |
| CrossVit-S | 81.0 | 5.6 | 1958.4 | 26.9 | 2.64 |
| Efficientformer-L3 | 82.4 | 3.9 | 2482.5 | 31.4 | 3.35 |
| PVTv2-B2 | 82.0 | 3.9 | 1848.3 | 25.4 | 2.49 |
| Isomorphic-Pruning4.2G | 82.4 | 4.2 | 2516.0 | 20.7 | 3.39 |
| HetDPT+-3.7G | 82.5 | 3.7 | 2763.9 | 20.0 | 3.73 |
| Isomorphic-Pruning2.6G | 81.1 | 2.6 | 3145.8 | 13.1 | 4.24 |
| HetDPT+-2.4G | 81.4 | 2.4 | 3845.6 | 13.2 | 5.19 |
| DeiT-T-KD | 74.5 | 1.3 | 6760.5 | 5.9 | 9.12 |
| NViT-T | 76.2 | 1.2 | 6013.7 | 6.7 | 8.11 |
| Isomorphic-Pruning1.2G | 77.5 | 1.2 | 6387.4 | 5.7 | 8.61 |
| HetDPT+-1.1G | 77.4 | 1.1 | 6814.0 | 6.2 | 9.19 |

We further benchmark our enhanced HetDPT+ against state-of-the-art extreme compression approaches, particularly width pruning methods such as Isomorphic Pruning and NViT, while also including results from efficient architecture designs like Swin, EfficientFormer, and CrossViT for broader context. As summarized in Table 9, HetDPT+ establishes new performance–efficiency trade-offs across different compression levels.

At moderate compression, HetDPT+-3.7G surpasses Isomorphic Pruning-4.2G in both accuracy (82.5% vs. 82.4%) and efficiency, reducing parameters (20.0M vs. 20.7M), lowering MACs (3.7G vs. 4.2G), and improving throughput (2763.9 vs. 2516.0), thereby increasing speedup to 3.77×.

At stronger compression levels, HetDPT+-2.4G achieves 81.4% accuracy compared to 81.1% with Isomorphic Pruning-2.6G, while delivering reduced compute cost and a faster 5.19× speedup, nearly 1× higher than the competitor.

Finally, in ultra-extreme settings, HetDPT+-1.1G delivers 77.4% accuracy using only 1.1G MACs, outperforming both DeiT-T and Isomorphic Pruning-1.2G, and reaching an impressive 9.19× speedup with throughput exceeding 6800 images/s on H800 PCIe.

These comprehensive results demonstrate that HetDPT+ pushes beyond the limitations of depth-width pruning, closing the gap with and even surpassing state-of-the-art width pruning methods, thus offering a new pathway towards extreme yet effective ViT compression.

**Excellent orthogonality to token reduction methods.** We further investigate the orthogonality between HetDPT's depth pruning and token reduction methods by combining HetDPT with both GTP-ViT ((Xu et al., 2024b)) and ToMe. Specifically, we apply each token reduction method to both Dense-DeiT-B and HetDPT, and compare their throughput–accuracy trade-offs in Fig. 8.

For GTP-ViT, HetDPT shows strong compatibility with token pruning across multiple operating points. For example, when the corresponding setting yields an accuracy of 81.47%, HetDPT+GTP achieves a throughput of 1461.76, corresponding to a 1.24× improvement over the dense baseline throughput of 1175.5, with only a 0.33% accuracy drop from the original 81.8%. More importantly, under comparable accuracy drops, HetDPT+GTP consistently remains above Dense-DeiT-B+GTP in throughput, indicating that the gains from depth pruning and token pruning are complementary rather than conflicting.

A similar trend is observed for ToMe. Across a wide range of operating points, HetDPT+ToMe consistently achieves higher throughput than Dense-DeiT-B+ToMe at similar accuracy levels. For instance, at around 0.9% accuracy drop, HetDPT+ToMe reaches a throughput of 1410.6, compared with 884.07 for Dense-DeiT-B+ToMe. Even at more aggressive token reduction settings, the advantage of HetDPT remains stable, showing that the benefit of depth pruning is preserved when combined with token merging.

Overall, these results demonstrate that the effectiveness of HetDPT is not limited to a single token pruning strategy or

a specific pruning ratio. Instead, HetDPT maintains strong orthogonality with both token pruning (GTP-ViT) and token merging (ToMe), consistently improving inference efficiency while preserving competitive accuracy.

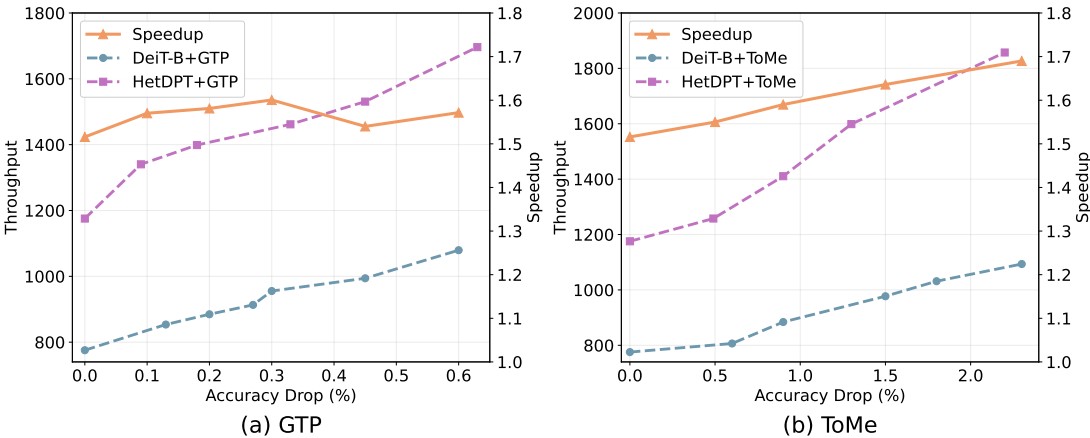

*Figure 8.* Orthogonality: HetDPT boosts inference efficiency by combining token pruning (a) GTP-ViT(b) ToMe.

## C. More Details About Polynomial Approximation of MAP

Roadmap. This appendix details: (i) the data collection procedures used to build the regression set for the MAP; (ii) the pre-transformation that maps discrete layer counts to normalized variables; (iii) the polynomial approximation of MAP with Leave-2-out cross-validation (L2OCV) for model selection; (iv) a concrete fitted result (degree, MAE/RMSE, and the polynomial); (v) the inverse transformation back to pruning ratios; (vi) runtime overhead; and (vii) a compact, colorized reference implementation.

### C.1. Data Collection

We employ two efficient data-collection procedures to obtain training samples for MAP regression: (1) single layer-type progressive pruning (Algorithm 1); and (2) interleaved pruning of attention and activation (Algorithm 2). Both procedures follow a prune–fine-tune–evaluate loop to produce representative samples at low cost.

---

**Algorithm 1:** Lightweight data collection for MAP regression (single layer type).

---

**Input:** pre-trained model $Y_0$, number of rounds $rds$, pruning budget $k$, total layers $L$

1   Initialize $train\_data = \{(\tilde{m}_0^a, \tilde{m}_0^g, a_0)\}$;

2   **Function** `PruneAlong` (*layer-type, $x_0$, $x_{\max}$*)**:**

3      **for** $n = 1$ **to** $rds$ **do**

        // Target pruning ratio along the specified layer type

4         $x_n = x_0 + n \cdot \frac{x_{\max} - x_0}{rds}$;

        // Prune the model

5         Prune $Y_{n-1}$ along *layer-type* to ratio $x_n$ to obtain $Y_n$;

        // Fine-tuning and evaluation

6         Fine-tune $Y_n$ and evaluate $\rightarrow (\tilde{m}_n^a, \tilde{m}_n^g, a_n)$;

7         Append $(\tilde{m}_n^a, \tilde{m}_n^g, a_n)$ to $train\_data$;

8      **end**

9   Set $\tilde{m}_{\max}^a = \frac{k}{L}$ and $\tilde{m}_{\max}^g = \frac{k}{L}$;

10   `PruneAlong` (*"attention"*, $\tilde{m}_0^a$, $\tilde{m}_{\max}^a$)**;**

11   `PruneAlong` (*"activation"*, $\tilde{m}_0^g$, $\tilde{m}_{\max}^g$)**;**

12   **return** $train\_data$

---

---

**Algorithm 2:** Lightweight data collection for MAP regression (interleaved pruning).

---

**Input:** pre-trained model $Y_0$, number of rounds $rds$, pruning budget $k$, total layers $L$

1   Initialize $train\_data = \{(\tilde{m}_0^a, \tilde{m}_0^g, a_0)\}$;

2   **Function** `PruneIterative`$((\tilde{m}_0^a, \tilde{m}_0^g), (\tilde{m}_{\max}^a, \tilde{m}_{\max}^g))$**:**

3     **for** $n = 1$ **to** $rds$ **do**

      `// Step 1: prune attention`

4       $\tilde{m}_n^a = \tilde{m}_0^a + n \cdot \frac{\tilde{m}_{\max}^a - \tilde{m}_0^a}{rds}$;

5       $\tilde{m}_{n-1}^g = \tilde{m}_0^g + (n - 1) \cdot \frac{\tilde{m}_{\max}^g - \tilde{m}_0^g}{rds}$;

6       Prune $Y_{n-1}$ along *attention* to ratio $\tilde{m}_n^a$ to obtain $Y_n'$;

      `// Fine-tuning and evaluation`

7       Fine-tune $Y_n'$ and evaluate $\rightarrow (\tilde{m}_n^a, \tilde{m}_{n-1}^g, a_n)$;

8       Append $(\tilde{m}_n^a, \tilde{m}_{n-1}^g, a_n)$ to $train\_data$;

      `// Step 2: prune activation`

9       $\tilde{m}_n^g = \tilde{m}_0^g + n \cdot \frac{\tilde{m}_{\max}^g - \tilde{m}_0^g}{rds}$;

10      Prune $Y_n'$ along *activation* to ratio $\tilde{m}_n^g$ to obtain $Y_n$;

      `// Fine-tuning and evaluation`

11      Fine-tune $Y_n$ and evaluate $\rightarrow (\tilde{m}_n^a, \tilde{m}_n^g, a_n)$;

12      Append $(\tilde{m}_n^a, \tilde{m}_n^g, a_n)$ to $train\_data$;

13    **end**

14   Set $\tilde{m}_{\max}^a = \frac{k}{L}$ and $\tilde{m}_{\max}^g = \frac{k}{L}$;

15   `PruneIterative`$((\tilde{m}_0^a, \tilde{m}_0^g), (\tilde{m}_{\max}^a, \tilde{m}_{\max}^g))$;

16   **return** $train\_data$

---

### C.2. Pre-Transformation from Discrete Layer Counts to Normalized Variables

The original pruning configuration is specified by discrete layer counts. Let $n_a$ and $n_g$ denote the numbers of *retained* attention and activation layers, respectively, and let $L$ be the total number of layers. We convert these to *retained ratios*

$$a := \frac{n_a}{L}, \qquad t := \frac{n_g}{L}, \qquad a, t \in \left\{ \frac{1}{L}, \frac{2}{L}, \ldots, 1 \right\}.$$

If the original description uses "prune $x$ layers", then the retained layer count is $n = L - x$, hence the retained ratio is $a = (L - x)/L$ (and similarly for $t$). This transformation not only facilitates smooth performance prediction through low-degree polynomial regression in a continuous domain, but also reformulates the discrete pruning budget into an analytically tractable linear constraint $(a + t = 2 - k/L)$ in the $(a, t)$ space, thereby simplifying theoretical analysis and enabling efficient optimization.

In the main text, $\tilde{m}^a$ and $\tilde{m}^g$ denote pruning ratios:

$$\tilde{m}^a = \frac{1}{L} \sum_{l=1}^{L} (1 - \hat{m}_l^a), \qquad \tilde{m}^g = \frac{1}{L} \sum_{l=1}^{L} (1 - \hat{m}_l^g).$$

In this appendix, we use retained ratios $(a, t)$ for polynomial fitting. They are related to the main-text pruning ratios by

$$a = 1 - \tilde{m}^a, \qquad t = 1 - \tilde{m}^g.$$

Therefore, the pruning-budget constraint $\tilde{m}^a + \tilde{m}^g = k/L$ is equivalent, in the $(a, t)$ domain, to

$$a + t = 2 - \frac{k}{L}.$$

In the empirical tables (e.g., Table 10), each $(a, t)$ pair already denotes the retained ratios.

## C.3. Polynomial Approximation and Leave-2-out Cross-Validation

We approximate MAP with a bivariate polynomial of total degree $\kappa$:

$$\mathcal{P}(a, t; \Theta) = \sum_{i+j \leq \kappa} \theta_{ij} \, a^i t^j.$$

To reduce overfitting given limited samples, $\kappa = 2$ is typically sufficient in practice. We perform Leave-2-out cross-validation (L2OCV) over $\kappa \in \{1, 2, 3, 4\}$: for each $\kappa$, we repeatedly leave out two samples, fit least squares on the remaining data, predict the held-out samples, aggregate predictions for all points, and compute MAE/RMSE. We then pick the $\kappa$ with the lowest validation RMSE, refit on the full dataset to obtain the final $\hat{\Theta}$, and search along the linear constraint $a + t = 2 - k/L$ over the discrete feasible set to find the recommended configuration.

With the dataset in Table 10 and the discrete constraint $a + t = 16/12$, L2OCV selects degree $\kappa = 2$ with

$$\text{MAE} = 0.4066, \quad \text{RMSE} = 0.4870.$$

The fitted polynomial MAP is

$$\hat{\mathcal{P}}(a, t) = 31.68 \,+\, 50.65 \, a \,+\, 39.29 \, t \,-\, 19.79 \, a^2 \,-\, 8.34 \, at \,-\, 11.70 \, t^2. \tag{8}$$

For completeness, the unrounded coefficients (from the implementation) are: 31.684374, 50.653461, 39.298158, $-19.795489$, $-8.338992$, $-11.704586$, corresponding to the terms $1$, $a$, $t$, $a^2$, $at$, and $t^2$ respectively.

An example of DeiT-Base with $L = 12$ is shown in Table 10, where $a$ and $t$ are retained ratios $(n/L)$:

*Table 10.* Accuracy under different retained ratios $(a, t)$.

| $a$ | $t$ | Accuracy (%) | $a$ | $t$ | Accuracy (%) |
|------|------|------|------|------|------|
| 1.00 | 1.00 | 81.80 | 1.00 | 0.92 | 81.07 |
| 0.92 | 1.00 | 81.31 | 1.00 | 0.83 | 80.40 |
| 0.83 | 1.00 | 80.90 | 1.00 | 0.75 | 78.90 |
| 0.75 | 1.00 | 80.20 | 1.00 | 0.67 | 77.80 |
| 0.67 | 1.00 | 78.10 | 1.00 | 0.58 | 76.70 |
| 0.58 | 1.00 | 77.40 | 1.00 | 0.50 | 75.20 |
| 0.50 | 1.00 | 72.69 | 1.00 | 0.42 | 74.50 |
| 0.42 | 1.00 | 69.44 | 1.00 | 0.33 | 73.90 |
| 0.33 | 1.00 | 64.84 | 0.92 | 0.92 | 80.34 |
| 0.92 | 0.83 | 80.03 | 0.83 | 0.83 | 78.88 |
| 0.83 | 0.75 | 78.08 | 0.75 | 0.75 | 76.52 |

## C.4. Inverse Transformation and Reporting

Once the polynomial MAP is fitted as

$$\hat{\mathcal{P}}(a, t) = 31.68 \,+\, 50.65 \, a \,+\, 39.29 \, t \,-\, 19.79 \, a^2 \,-\, 8.34 \, at \,-\, 11.70 \, t^2, \tag{9}$$

we can express it in terms of the pruning ratios by the inverse transformation

$$\tilde{m}^a = 1 - a, \qquad \tilde{m}^g = 1 - t.$$

This yields the equivalent polynomial

$$\mathcal{P}(\tilde{m}^a, \tilde{m}^g) = 81.79 \,-\, 2.73 \, \tilde{m}^a \,-\, 7.55 \, \tilde{m}^g \,-\, 19.79 \, (\tilde{m}^a)^2 \,-\, 8.34 \, \tilde{m}^a \tilde{m}^g \,-\, 11.70 \, (\tilde{m}^g)^2. \tag{10}$$

The pruning budget requires

$$\tilde{m}^a + \tilde{m}^g = \frac{k}{L}.$$

Since both $\tilde{m}^a$ and $\tilde{m}^g$ take values from the finite discrete set $\{0, 1/L, \ldots, 1\}$ and must satisfy the budget constraint, the optimal configuration can be obtained simply by enumerating all feasible pairs $(\tilde{m}^a, \tilde{m}^g)$ and selecting the one with the highest value of $\mathcal{P}(\tilde{m}^a, \tilde{m}^g)$. This exhaustive search is computationally inexpensive given the small search space.

---

**Algorithm 3:** Exhaustive search for optimal pruning ratios.

---

**Input:** Pruning budget $k/L$, polynomial $\mathcal{P}(\tilde{m}^a, \tilde{m}^g)$
**Output:** Optimal configuration $(\tilde{m}^{a\star}, \tilde{m}^{g\star})$

1   best_score $\leftarrow -\infty$;
2   **for** $\tilde{m}^a \in \{0, 1/L, \ldots, k/L\}$ **do**
3      $\tilde{m}^g \leftarrow k/L - \tilde{m}^a$;
4      **if** $\tilde{m}^g \in \{0, 1/L, \ldots, 1\}$ **then**
5         score $\leftarrow \mathcal{P}(\tilde{m}^a, \tilde{m}^g)$;
6         **if** *score > best_score* **then**
7            best_score $\leftarrow$ score;
8            $(\tilde{m}^{a\star}, \tilde{m}^{g\star}) \leftarrow (\tilde{m}^a, \tilde{m}^g)$;
9         **end**
10      **end**
11 **end**
12 **return** $((\tilde{m}^a)^\star, (\tilde{m}^g)^\star)$;

---

Alternatively, one may directly optimize the polynomial in the retained-variable space $(a, t)$ along the line constraint $a + t = 2 - k/L$. The solution $(a^\star, t^\star)$ is then mapped back to pruning ratios via

$$(\tilde{m}^a)^\star = 1 - a^\star, \qquad (\tilde{m}^g)^\star = 1 - t^\star,$$

which automatically satisfies the budget. Both approaches are mathematically equivalent, as they differ only by a linear change of variables.

### C.5. Runtime

The MAP fitting is a small-scale least-squares regression on a handful of samples and low-degree polynomials. Its computational overhead is negligible compared to a single fine-tuning run. In practice, the MAP fitting adds virtually no runtime overhead to our pipeline.

### C.6. Reference Implementation

We provide a compact, colorized reference implementation that performs L2OCV to select the polynomial degree, fits the final model, and searches the discrete feasible set. Inputs use retained ratios $(a, t)$; to report pruning ratios, apply $\tilde{m}^a = 1 - a$ and $\tilde{m}^g = 1 - t$.

```python
import numpy as np
from itertools import combinations

TOTAL_SUM_NUM = 16   # a + t = TOTAL_SUM_NUM / 12

data = np.array([
    (1.00, 1.00, 81.8), (0.92, 1.00, 81.31), (0.83, 1.00, 80.9),
    (0.75, 1.00, 80.2), (0.67, 1.00, 78.2), (0.58, 1.00, 77.4),
```

```
9         (1.00, 0.92, 81.26), (1.00, 0.83, 80.4), (1.00, 0.75, 78.9),
10        (1.00, 0.67, 77.8), (1.00, 0.58, 76.7),
11        (0.92, 0.92, 80.34), (0.92, 0.83, 80.03),
12        (0.83, 0.83, 78.88), (0.83, 0.75, 78.08),
13        (0.75, 0.75, 76.52),
14    ], dtype=float)
15
16    A, y = data[:, :2], data[:, 2]
17
18    def exps(k): return [(i, t-i) for t in range(k+1) for i in range(t+1)]
19    def Phi(A, k):
20        E = exps(k)
21        return np.stack([np.prod(np.power(x, E), axis=1) for x in A], 0)
22    def fit(X, y): return np.linalg.lstsq(X, y, rcond=None)[0]
23    def pred(X, c): return X @ c
24    def l2ocv(A, y, k):
25        N = len(y); P = np.zeros(N); C = np.zeros(N, int)
26        for i, j in combinations(range(N), 2):
27            m = np.ones(N, bool); m[[i, j]] = False
28            c = fit(Phi(A[m], k), y[m])
29            for t in (i, j):
30                P[t] += pred(Phi(A[[t]], k), c)[0]; C[t] += 1
31        e = P/C - y
32        return np.mean(np.abs(e)), np.sqrt(np.mean(e**2))
33    def search_best(c, k, total=TOTAL_SUM_NUM):
34        kmin, kmax = max(1, total-11), min(11, total-1)
35        cand = []
36        for s in range(kmin, kmax+1):
37            a, t = s/12, (total-s)/12
38            acc = float(pred(Phi(np.array([[a, t]]), k), c)[0])
39            cand.append((s, a, t, acc))
40        return max(cand, key=lambda z: z[3]), cand
41
42    results = []
43    for deg in [1, 2, 3, 4]:
44        mae, rmse = l2ocv(A, y, deg)
45        C = fit(Phi(A, deg), y)
46        best, _ = search_best(C, deg)
47        results.append((deg, mae, rmse, C, best))
48    best_res = min(results, key=lambda r: r[2])
49    deg, mae, rmse, C, (s, a, t, acc) = best_res
50    print(f"Best degree: {deg}, MAE={mae:.4f}, RMSE={rmse:.4f}")
51    print(f"Recommended: a={a:.4f}, t={t:.4f}, pred acc={acc:.4f}")
52    print(f"Pruning ratios: tau_ma={1-a:.4f}, tau_mg={1-t:.4f}")
```

## D. More Details About Ablation Study

In the ablation study, we conduct experiments on the two steps of Stage 1 in HetDPT. Additionally, we compare the effects of applying TE-Static and individual components for heterogeneous pruning. This section explicitly lists the pruning indices to better illustrate the challenges mentioned in our insights.

**Without TE-Static and Stage1-Step1** We initially evaluated a baseline pruning strategy that directly selects layers using trainable importance parameters without TE-Static and Stage1-Step1. This approach methodologically neglects the gradient disparity challenge between self-attention layers and activation layers, instead employing direct cross-type comparisons. As demonstrated in Tab.11, this method consistently induced systematic performance degradation across various architectures (DeiT-B and DeiT-S) and pruning intensities. Further analysis reveals its bias towards pruning more activation layers, likely due to inherent gradient magnitude disparities between layer types.

**Without Stage1-Step2 (Trainable importance scores)** We subsequently validated the critical role of Stage1-Step2 through a comparative approach that directly uses TE-Static for layer selection without trainable importance scores. This method fails to model layer interdependencies and eliminate cross-type gradient disparity challenges. As shown in Tab. 12, this

method exhibits significantly degraded performance compared to HetDPT, conclusively demonstrating the necessity of Stage1-Step2's trainable importance mechanism.

*Table 11.* Ablation on TE-Static and Stage1-Step1: HetDPT (bold) vs. without TE-Static and Stage1-Step1 (non-bold)

| Models | Pruned | Self-attention Index | Activation Index | Acc |
|---|---|---|---|---|
| DeiT-S | 6 layers | [7,11]
**[1,10,11]** | [0,1,10,11]
**[8,10,11]** | 79.6
**80.1** |
| | 8 layers | [7,8,11]
**[1,7,10,11]** | [0,1,9,10,11]
**[7,8,10,11]** | 78.5
**79.7** |
| DeiT-B | 8 layers | [0,3]
**[0,3,7,11]** | [0,1,3,4,9,10]
**[2,7,8,11]** | 80.8
**82.1** |
| | 10 layers | [0,1,2,5]
**[0,3,7,8,11]** | [0,1,2,3,9,10]
**[2,7,8,10,11]** | 78.1
**81.8** |

*Table 12.* Ablation on Stage1-Step2: HetDPT (bold) vs. without trainable importance scores (non-bold)

| Models | Pruned | Self-attention Index | Activation Index | Acc |
|---|---|---|---|---|
| DeiT-S | 6 layers | [1,9,11]
**[1,10,11]** | [9,10,11]
**[8,10,11]** | 79.9
**80.1** |
| | 8 layers | [1,9,10,11]
**[1,7,10,11]** | [8,9,10,11]
**[7,8,10,11]** | 79.4
**79.7** |
| DeiT-B | 8 layers | [0,3,9,11]
**[0,3,7,11]** | [7,8,9,10]
**[2,7,8,11]** | 81.1
**82.1** |
| | 10 layers | [0,3,8,9,11]
**[0,3,7,8,11]** | [7,8,9,10,11]
**[2,7,8,10,11]** | 80.5
**81.8** |

*Table 13.* HetDPT (bold) vs. TE-Static for heterogeneous depth pruning (non-bold)

| Models | Pruned | Self-attention Index | Activation Index | Acc |
|---|---|---|---|---|
| DeiT-S | 6 layers | [1,9,10,11]
**[1,10,11]** | [9,10]
**[8,10,11]** | 79.7
**80.1** |
| | 8 layers | [1,8,9,10,11]
**[1,7,10,11]** | [9,10,11]
**[7,8,10,11]** | 78.5
**79.7** |
| DeiT-B | 8 layers | [5,7,9,10,11]
**[0,3,7,11]** | [6,10,11]
**[2,7,8,11]** | 79.4
**82.1** |
| | 10 layers | [5,7,9,10,11]
**[0,3,7,8,11]** | [5,6,9,10,11]
**[2,7,8,10,11]** | 78.9
**81.8** |

**TE-Static for heterogeneous pruning** Finally, we investigated a comparative method that directly employs TE-Static for heterogeneous depth pruning without any training components. This approach calculates the TE metric for both self-attention layers and activation layers, subsequently selecting layers with minimal TE scores as pruning candidates. As demonstrated in Table 13, the TE-Static heterogeneous depth pruning exhibits significant performance degradation. Notably, in contrast to the learning-based pruning observed in Table 11, this method disproportionately removes self-attention layers. This observation aligns with the Recovery Asymmetry challenge discussed before, where self-attention layers demonstrate weaker accuracy recovery capacity during iterative pruning compared to activation layers.

## E. Visualization

We employ Attention Explanation (Chefer et al., 2021) to project the attention maps from the output token back onto the input image, thereby enabling the visualization and interpretation of different blocks of multi-head self-attention within DeiT-Base. This approach provides a fine-grained perspective on how information is propagated and aggregated across layers, and allows us to explicitly observe which regions of the image are emphasized by different attention heads and blocks.

As illustrated in Figure 9 and Figure 10, our method identifies and automatically prunes attention layers that are redundant. Taking the results in Figure 9 as an example, the layers removed include Layer 0, Layer 3, Layer 7, and Layer 11. A closer examination shows that these pruned layers share highly similar attention distributions with their neighboring layers.

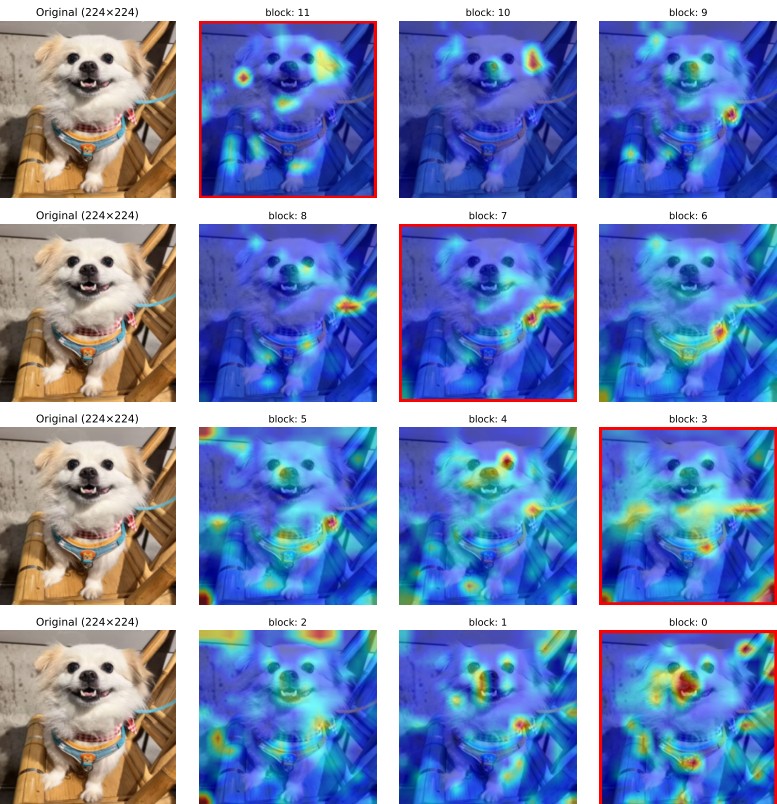

*Figure 9.* Visualization of cute dog: input image (left) and visualization of attention maps (right) of DeiT-Base. Attention maps with red bounding boxes are the attention layers to be removed.

Specifically, Layer 0 and Layer 1 both focus on the global context of the dog and its surrounding background; Layers 3, 4, and 5 collectively attend to the dog's body; Layers 6, 7, and 8 capture similar details of the clothing; and Layers 10 and 11 concentrate on the facial region. Such patterns suggest that the removed layers contribute little additional information beyond what is already captured by adjacent layers.

A similar trend can be observed in Figure 10 for the cat example, where the discarded layers also exhibit overlapping attention with those retained. This consistent redundancy across different samples underlines the robustness of our pruning strategy. Beyond its practical effect in reducing model complexity, this phenomenon also offers interpretability insights: it indicates that not all attention blocks within transformer architectures contribute equally to the final representation, and some primarily replicate context already modeled by their neighbors.

Collectively, these findings highlight that our approach not only suppresses unnecessary redundancy within the attention structure but also enhances the efficiency and interpretability of the model. By demonstrating that functionally overlapping layers can be safely pruned without sacrificing focus on key semantic regions, our method indirectly validates the hypothesis that a more compact layer set can maintain — and potentially improve — the effectiveness of transformer-based vision models.

## F. Theoretical Foundations of MAP

This appendix provides a strengthened and fully rigorous theoretical foundation for **Model Accuracy Predictor (MAP)**. Relative to the main text, we supply additional lemmas on the discreteness, continuity, polynomial approximability, and consistency under noise. The results consolidate and mathematically justify Theorem 1–3. Throughout, let $L$ denote the number of Transformer blocks.

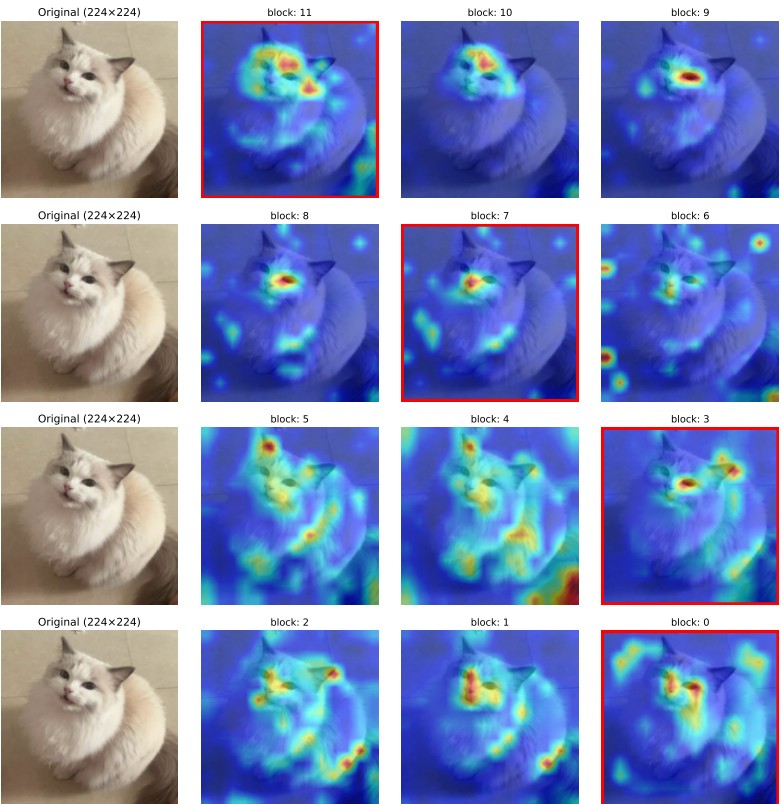

*Figure 10.* Visualization of cute cat: input image (left) and visualization of attention maps (right) of DeiT-Base. Attention maps with red bounding boxes are the attention layers to be removed.

### F.1. Preliminaries

Assume a ViT $\mathcal{F}$ has $L$ Transformer blocks. The $l$-th Transformer block $f_l$ typically integrates: 1) two linear layers, i.e., $\mathcal{L}_{l,1}$ and $\mathcal{L}_{l,2}$, 2) an attention layer $\mathcal{A}_l$, 3) a GELU activation function layer $\mathcal{G}_l$. For simplicity, we omit auxiliary components such as residual connections and layer normalization. Thus,

$$f_l(\cdot) := \mathcal{L}_{l,2} \circ \mathcal{G}_l \circ \mathcal{L}_{l,1} \circ \mathcal{A}_l \circ f_{l-1}(\cdot), \tag{11}$$

where $\circ$ denotes the function composition operation. We aim to reduce computational overhead through heterogeneous depth pruning of attention layers and redundant GELU activation function layers. That is,

$$\begin{aligned}
f_l^{\hat{m}}(\cdot) &:= \mathcal{L}_{l,2} \circ h\left(\mathcal{G}_l\right) \circ \mathcal{L}_{l,1} \circ h\left(\mathcal{A}_l\right) \circ f_{l-1}^{\hat{m}}(\cdot), \\
h\left(\mathcal{G}_l\right) &= \hat{m}_l^g \circ \mathcal{G}_l + (1 - \hat{m}_l^g) \circ I, \\
h\left(\mathcal{A}_l\right) &= \hat{m}_l^a \circ \mathcal{A}_l + (1 - \hat{m}_l^a) \circ I, \\
\hat{m}_l^a, &\hat{m}_l^g \in \{0,1\}. \\
M &:= \left(\{\hat{m}_l^a\}_{l=1}^L, \{\hat{m}_l^g\}_{l=1}^L\right)
\end{aligned} \tag{12}$$

Notably, $\hat{m}_l^g$ and $\hat{m}_l^a$ are the binary masks. A mask's value of 1 preserves the corresponding layer, while 0 prunes it. $I$ denotes the identity mapping, i.e., a direct skip connection when the layer is pruned. $M$ is the set of all the binary masks. A ViT with such masks is referred to as $\mathcal{F}^{\hat{m}}$. Thus, the heterogeneous depth pruning problem is formulated as:

$$\begin{aligned}
\underset{W,M}{\arg\min} \; &\ell\left(\mathcal{F}^{\hat{m}}(X, W, M), Y\right), \\
&\text{s.t.} \quad \tilde{m}^a + \tilde{m}^g = k/L, \\
\tilde{m}^a = \frac{1}{L}\sum_{l=1}^L \mathbf{1}\{\hat{m}_l^a = 0\}, \quad &\tilde{m}^g = \frac{1}{L}\sum_{l=1}^L \mathbf{1}\{\hat{m}_l^g = 0\}.
\end{aligned} \tag{13}$$

where $(X, Y)$ are respectively samples and labels of fitting data, and $W$ is the weights of $\mathcal{F}^{\hat{m}}$. $\ell(\cdot)$ refers to the loss function to measure data fitting quality, e.g., cross-entropy loss for classification tasks. Thus the admissible ratio set $\mathcal{D}$ is

$$\mathcal{D} = \{0, \tfrac{1}{L}, \ldots, 1\}^2, \qquad |\mathcal{D}| = (L+1)^2.$$

For each $(\tilde{m}^a, \tilde{m}^g) \in \mathcal{D}$, the prune–finetune–evaluate pipeline yields a deterministic validation accuracy $\mathcal{P}$:

$$\mathcal{P} : \mathcal{D} \to [0, 1].$$

## F.2. Well-Posedness of Pruning–Ratio Optimization (Full Proof for Theorem 3.1)

We collect and strengthen the results establishing that an optimal pruning configuration always exists under any pruning budget.

**Lemma F.1** (Finite representability). *$\mathcal{D}$ is a finite subset of $[0, 1]^2$.*

*Proof.* Immediate from $\mathcal{D} = \{(i/L, j/L) : 0 \le i, j \le L\}$. □

**Lemma F.2** (Functional well-definedness). *$\mathcal{P} : \mathcal{D} \to [0, 1]$ is well-defined.*

*Proof.* Each point in $\mathcal{D}$ corresponds to binary masks $\hat{m}^a, \hat{m}^g$. The pipeline is deterministic and outputs a single accuracy value. □

A pruning budget imposes

$$\tilde{m}^a + \tilde{m}^g = k/L,$$

yielding the constrained feasible set $\mathcal{D}_k$:

$$\mathcal{D}_k = \left\{ (\tilde{m}^a, \tilde{m}^g) \in \mathcal{D} : \tilde{m}^a + \tilde{m}^g = k/L \right\}.$$

**Lemma F.3** (Feasible set is finite). *$|\mathcal{D}_k| \le L + 1$.*

*Proof.* If $\tilde{m}^a = i/L$, then $\tilde{m}^g = (k-i)/L$, hence at most $L+1$ feasible pairs. □

**Lemma F.4** (Attainment of maximum over a finite domain). *If set $S$ is finite and $f : S \to [0, 1]$, then $\max_{x \in S} f(x)$ exists.*

*Proof.* A finite set has finitely many function values, whose maximum exists. □

We now restate and prove the main theorem.

**Theorem F.5** (Restatement of Theorem 3.1). *For any pruning budget $k$, there exists an optimal pruning configuration:*

$$(\tilde{m}^{a\star}, \tilde{m}^{g\star}) \in \underset{(\tilde{m}^a, \tilde{m}^g) \in \mathcal{D}_k}{\arg\max} \; \mathcal{P}(\tilde{m}^a, \tilde{m}^g)$$

*Proof.* By Lemma F.3, $\mathcal{D}_k$ is finite. By Lemma F.2, $\mathcal{P}$ is well-defined on $\mathcal{D}_k$. The result then follows from Lemma F.4. $\square$

This establishes that pruning-ratio optimization is **well-posed**: for any budget constraint, an optimal solution is guaranteed to exist.

### F.3. Polynomial Approximation of the Accuracy Surface (Full Proof for Theorem 3.3)

This section provides the complete proof of Theorem 3.3, establishing that the pruning-accuracy surface admits arbitrarily accurate polynomial approximations.

RELAXATION ASSUMPTION

**Assumption F.6** (Continuous Relaxation of Pruning Ratios). Although structured pruning masks $(\hat{m}^a, \hat{m}^g)$ are discrete, their normalized pruning ratios
$$(\tilde{m}^a, \tilde{m}^g) \in \mathcal{D} = \{0, \tfrac{1}{L}, \ldots, 1\}^2 \subset [0, 1]^2$$
admit a continuous relaxation in which $(\tilde{m}^a, \tilde{m}^g)$ is treated as a point in the full square $[0, 1]^2$. Furthermore, the accuracy functional $\mathcal{P}$ evaluated on $\mathcal{D}$ extends to a continuous map
$$\tilde{\mathcal{P}} : [0, 1]^2 \to [0, 1], \qquad \tilde{\mathcal{P}}|_{\mathcal{D}} = \mathcal{P}.$$

**Justification.** This assumption is standard in pruning theory and inherits two empirical/structural facts:

1. *Sparsity relaxations.* Continuous formulations of pruning (e.g., magnitude relaxation, soft masks) induce validation accuracy that varies smoothly with pruning level.

2. *Observed one-dimensional smoothness.* Accuracy under $(\tilde{m}^a, 0)$ or $(0, \tilde{m}^g)$ varies smoothly with pruning ratio, suggesting that the bivariate surface is continuous.

Assumption F.6 therefore formalizes a widely accepted—and empirically validated— interpretation of pruning ratios as living on a continuous compact domain.

SUPPORTING LEMMAS

**Lemma F.7** (Domain compactness). *The domain $[0, 1]^2$ is a compact Hausdorff space.*

*Proof.* Each interval $[0, 1]$ is compact Hausdorff; products preserve both properties. $\square$

**Lemma F.8** (Polynomial algebra as separating lattice). *Let $A = \mathbb{R}[x, y]$ denote the algebra of all real bivariate polynomials. Then:*

1. *$A$ contains all constant functions;*

2. *$A$ separates points of $[0, 1]^2$;*

3. *The uniform closure $\overline{A}$ is closed under $\max$ and $\min$, i.e.,*

$$\max(f, g), \; \min(f, g) \in \overline{A}, \qquad \forall f, g \in \overline{A}.$$

*Hence $A$ is a* separating lattice.

*Proof.* (1) Constant polynomials belong to $A$.

(2) If $(x_1, y_1) \neq (x_2, y_2)$, then either $x_1 \neq x_2$ or $y_1 \neq y_2$. Thus $p(x, y) = x$ or $p(x, y) = y$ separates the points.

(3) For any $f, g \in \overline{A}$,

$$\max(f, g) = \frac{f + g}{2} + \frac{|f - g|}{2}, \qquad \min(f, g) = \frac{f + g}{2} - \frac{|f - g|}{2}.$$

Since absolute value $|\cdot|$ is continuous and can be uniformly approximated on compact sets by polynomials, $|f - g| \in \overline{A}$, completing the proof. $\square$

**Lemma F.9** (Stone–Weierstrass theorem, lattice version). *Let $K$ be a compact Hausdorff space and $A \subset C(K)$ a separating lattice containing constants. Then $\overline{A} = C(K)$ under the uniform norm.*

MAIN RESULT

**Theorem F.10** (Restatement of Theorem 3.3). *Under Assumption F.6, for every $\varepsilon > 0$ there exists a polynomial $Q(x, y) \in \mathbb{R}[x, y]$ such that*

$$\sup_{(x,y) \in [0,1]^2} \left| \tilde{\mathcal{P}}(x, y) - Q(x, y) \right| < \varepsilon.$$

*Consequently,*

$$\max_{(\tilde{m}^a, \tilde{m}^g) \in \mathcal{D}} \left| \mathcal{P}(\tilde{m}^a, \tilde{m}^g) - Q(\tilde{m}^a, \tilde{m}^g) \right| < \varepsilon.$$

*Proof.* By Lemma F.7, $[0, 1]^2$ is compact Hausdorff. By Lemma F.8, the polynomial algebra $A$ is a separating lattice containing constants. Thus all hypotheses of the Stone–Weierstrass theorem (lattice version), Lemma F.9, are satisfied.

Therefore $\overline{A} = C([0, 1]^2)$, meaning that polynomials are uniformly dense in the space of continuous functions on the domain. Since $\tilde{\mathcal{P}} \in C([0, 1]^2)$ by Assumption F.6, there exists a polynomial $Q$ such that

$$\sup_{(x,y) \in [0,1]^2} |\tilde{\mathcal{P}}(x, y) - Q(x, y)| < \varepsilon.$$

Because $\mathcal{D} \subset [0, 1]^2$, the same inequality holds when restricted to $\mathcal{D}$. $\square$

REMARK

Theorem F.10 guarantees that the empirical pruning-accuracy landscape can be approximated to arbitrary precision by finite-degree bivariate polynomials. This result provides the theoretical foundation for polynomial-based surrogate models in pruning-ratio optimization, including the MAP framework proposed in this work.

### F.4. Consistency of MAP Under Noisy Fast Finetuning (Full Proof of Theorem 3.5)

This section provides a rigorous statistical justification for the MAP when observed accuracies are obtained via subset-based fast finetuning. We show that the noise introduced by fast finetuning becomes an irreducible constant in the least squares loss and does not affect the identification of optimal pruning configurations.

OBSERVATION MODEL AND ASSUMPTIONS

**Assumption F.11** (Noisy observation model). For any pruning configuration $x = (\tilde{m}^a, \tilde{m}^g) \in \mathcal{D}$, let $\mathcal{P}(x)$ denote the ground-truth validation accuracy under full finetuning. The subset-based fast finetuning produces observations

$$y = \mathcal{P}(x) + b + \varepsilon, \tag{14}$$

where:

1. $b \in \mathbb{R}$ is a constant bias independent of $x$, arising from the limited number of finetuning steps;

2. $\varepsilon$ is a zero-mean random variable with finite variance $\mathrm{Var}(\varepsilon) = \sigma^2 < \infty$;

3. $\varepsilon$ is independent of $x$.

The independence assumption is justified by the fact that the randomness in fast finetuning stems from subset sampling and optimization stochasticity, which are independent of the pruning configuration.

INVARIANCE OF MAXIMIZER UNDER CONSTANT BIAS

**Lemma F.12** (Bias-invariance of maximizers). *For any objective function $\mathcal{P}(x)$ and a constant $b$, the set of maximizers remains unchanged:*

$$\arg\max_x \mathcal{P}(x) = \arg\max_x \left( \mathcal{P}(x) + b \right).$$

*Proof.* For any $x_1, x_2 \in X$, we have $\mathcal{P}(x_1) \geq \mathcal{P}(x_2)$ if and only if $\mathcal{P}(x_1) + b \geq \mathcal{P}(x_2) + b$. Thus the ordering is preserved, and the set of maximizers remains unchanged. $\square$

LEAST SQUARES DECOMPOSITION

Let $f(x; \theta)$ denote the MAP predictor parameterized by $\theta$. The population risk under squared error loss is

$$\mathcal{L}(\theta) = \mathbb{E}_{x,\varepsilon}\left[ (y - f(x; \theta))^2 \right].$$

**Lemma F.13** (Bias-variance decomposition). *Under Assumption F.11, the population risk decomposes as*

$$\mathcal{L}(\theta) = \mathbb{E}_x\left[ (\mathcal{P}(x) + b - f(x; \theta))^2 \right] + \sigma^2. \tag{15}$$

*Proof.* Substituting the observation model (14) and letting $A(x; \theta) := \mathcal{P}(x) + b - f(x; \theta)$, we obtain

$$\mathcal{L}(\theta) = \mathbb{E}_{x,\varepsilon}\left[ (A(x; \theta) + \varepsilon)^2 \right] = \mathbb{E}_{x,\varepsilon}\left[ A(x; \theta)^2 + 2A(x; \theta)\varepsilon + \varepsilon^2 \right].$$

By linearity of expectation and the independence of $\varepsilon$ from $x$,

$$\mathbb{E}_{x,\varepsilon}[A(x; \theta) \cdot \varepsilon] = \mathbb{E}_x[A(x; \theta)] \cdot \mathbb{E}[\varepsilon] = 0,$$

since $\mathbb{E}[\varepsilon] = 0$. The result follows from $\mathbb{E}[\varepsilon^2] = \sigma^2$. $\square$

MAIN RESULT

**Theorem F.14** (Consistency of MAP under noisy observations, restatement of Theorem 3.5). *Under Assumptions F.6 and F.11, suppose the MAP predictor $f(x; \theta)$ is trained by minimizing the least squares loss over $N$ i.i.d. samples. Then:*

1. *The noise variance $\sigma^2$ does not affect the gradient $\nabla_\theta \mathcal{L}(\theta)$;*

2. *As $N \to \infty$, the optimal predictor satisfies $f^*(x) = \mathcal{P}(x) + b$;*

3. *The maximizer of $f^*$ coincides with that of $\mathcal{P}$:* $\arg\max_x f^*(x) = \arg\max_x \mathcal{P}(x)$.

*Proof.* *Statement 1.* From Lemma F.13, we have $\mathcal{L}(\theta) = \mathcal{L}_{\text{clean}}(\theta) + \sigma^2$, where $\mathcal{L}_{\text{clean}}(\theta) := \mathbb{E}_x[(\mathcal{P}(x) + b - f(x; \theta))^2]$. Since $\sigma^2$ is constant with respect to $\theta$,

$$\nabla_\theta \mathcal{L}(\theta) = \nabla_\theta \mathcal{L}_{\text{clean}}(\theta).$$

*Statement 2.* The minimizer of the population squared error loss is the conditional expectation of the target. Under Assumption F.11,

$$f^*(x) = \mathbb{E}[y \mid x] = \mathbb{E}[\mathcal{P}(x) + b + \varepsilon \mid x] = \mathcal{P}(x) + b,$$

where the last equality uses the fact that $\mathcal{P}(x)$ and $b$ are deterministic given $x$, and $\mathbb{E}[\varepsilon] = 0$. By Theorem F.10, polynomials can approximate $\mathcal{P}(x) + b$ to arbitrary precision, ensuring that the MAP model class is sufficiently expressive.

*Statement 3.* This follows immediately from Lemma F.12 with $f^*(x) = \mathcal{P}(x) + b$. $\square$

REMARK

The decomposition in Lemma F.13 reveals that the noise variance $\sigma^2$ acts as an irreducible error floor that does not influence the optimization landscape. The constant bias $b$ shifts the predictor uniformly but preserves the ranking of configurations. Consequently, MAP trained via least squares remains a consistent estimator for identifying optimal pruning ratios, even when observations are corrupted by the approximations inherent in fast finetuning.

### F.5. Theoretical Foundation of MAP: A Unified View

Theorems F.5–F.14 collectively establish a rigorous mathematical foundation for the MAP framework. The three results form a logical chain that addresses existence, approximation, and robustness, respectively.

**Existence (Theorem F.5).** The pruning-ratio optimization problem is well-posed. Since the feasible set $\mathcal{D}_k$ is finite and the accuracy functional $\mathcal{P}$ is bounded, a global maximizer $(\tilde{m}^{a\star}, \tilde{m}^{g\star})$ is guaranteed to exist for any budget constraint. This foundational result ensures that the optimization target is mathematically meaningful.

**Approximability (Theorem F.10).** The accuracy surface $\mathcal{P}$ admits arbitrarily accurate polynomial approximations. By invoking the Stone–Weierstrass theorem under Assumption F.6, we establish that bivariate polynomials are dense in the space of continuous functions on the pruning-ratio domain. This justifies the use of polynomial regressors in MAP and guarantees sufficient expressive power to capture the underlying accuracy landscape.

**Consistency (Theorem F.14).** MAP remains a consistent estimator even when trained on noisy observations from fast finetuning. The least squares objective decomposes such that the noise variance $\sigma^2$ contributes only an irreducible constant, leaving the optimization gradient unchanged. Meanwhile, the systematic bias $b$ uniformly shifts the predictor without altering the ranking of configurations. Consequently, the maximizer of the learned predictor coincides with that of the ground-truth accuracy surface.

**Unified Guarantee.** The three theorems together ensure that MAP searches for a *guaranteed-to-exist* optimum (Theorem F.5) using a *provably expressive* approximator (Theorem F.10) that *consistently recovers* the true maximizer despite being trained on low-cost, noisy proxies (Theorem F.14). This theoretical triad provides a complete justification for the practical effectiveness of the MAP framework.

