# OpenReview forum: "Rethinking Depth Pruning for Vision Transformers: A Heterogeneity-Aware Perspective"
_ICML.cc/2026/Conference — ICML 2026 regular_

### Official Review · Reviewer_xtD9 · 2026-03-09

**Soundness:** 3
**Presentation:** 3
**Significance:** 3
**Originality:** 3
**Overall Recommendation:** 4
**Confidence:** 4

**Summary:**

This paper attributes ViT depth pruning degradation to the heterogeneity between attention and activation layers. To address this, the authors propose HetDPT, which uses a Model Accuracy Predictor (MAP) to optimally allocate pruning budgets. By removing redundant activation layers and merging adjacent linear layers, HetDPT resolves dimension mismatch and achieves state-of-the-art inference acceleration with near-lossless accuracy.

**Compliance With Llm Reviewing Policy:**

Affirmed.

**Final Justification:**

The authors have appropriately answered my questions and resolved my doubts, so I will maintain my positive score.

**Key Questions For Authors:**

1. Regarding the limited expressivity of the low-degree polynomial: Although the theoretical analysis proves approximability to arbitrary precision, is there a practical risk that a simple quadratic surface might underfit the complex, high-dimensional interactions between attention and activation layers, thereby leading to suboptimal pruning budgets?
2. Dropping activation layers fundamentally removes critical non-linearities from the network. While performance on standard benchmarks like ImageNet-1k is maintained, does this reduction in non-linear operations intrinsically bottleneck the model's representational capacity for tasks that require highly complex, fine-grained feature extraction?

**Limitations:**

yes

**Strengths And Weaknesses:**

Strengths:
1. The approach of resolving the persistent dimension mismatch issue in depth pruning by removing the activation function and seamlessly merging adjacent linear layers is both practical and elegant.
2. The proposed method demonstrates excellent synergy when combined with existing token pruning techniques like GTP-ViT. Achieving an additional speedup at the cost of only a marginal accuracy drop substantially highlights its value for real-world deployment scenarios.

Weaknesses:
1. The core idea of removing activation functions solely to facilitate the merging of adjacent linear layers sacrifices the network's non-linearity. Consequently, this approach may be perceived more as a heuristic engineering trick rather than a fundamental architectural improvement.
2. While the authors theoretically prove that polynomial approximation can achieve arbitrary accuracy, the practical implementation relies on a low-degree polynomial to prevent overfitting. It remains questionable whether such a low-degree polynomial can accurately model the highly complex accuracy surfaces of extremely deep architectures.

---

> ### Author Rebuttal · Authors · 2026-03-31
>
> Thank you for the thoughtful questions and for recognizing our work. We appreciate these concerns, as they help us clarify both the practical design of MAP and the role of activation pruning in HetDPT.
>
> ---
>
> **1) A low-degree polynomial may be too simple to model the interaction between attention and activation pruning, especially for complex or extremely deep architectures.**
>
> In principle, an overly simple surrogate could underfit a complex pruning-performance landscape. However, in our method the polynomial degree is **not fixed a priori**. Instead, MAP selects the degree **empirically** using **Leave-2-out cross-validation (L2OCV)** over candidate degrees $\kappa \in \{1,2,3,4\}$. We will clarify this point more explicitly in the revision. The full MAP fitting procedure, data collection, cross-validation protocol, fitted polynomial, and exhaustive search process are provided in **Appendix C**.
>
> Concretely, MAP is fitted on a **small 2D budget space** parameterized only by the global retained ratios of attention and activation layers, rather than on a high-dimensional space over all layer identities. After collecting sample points via lightweight pruning/fine-tuning/evaluation, we fit candidate polynomials, evaluate them by L2OCV, and then select the degree with the lowest validation RMSE. For the DeiT-B example in **Appendix C**, L2OCV selects a **quadratic** model with **MAE = 0.4066** and **RMSE = 0.4870**, indicating that this low-degree surrogate already predicts the accuracy surface sufficiently well for pruning-budget allocation. The final budget is then chosen by **exhaustive search over the discrete feasible set**, so the polynomial serves only as a compact surrogate on a low-dimensional domain.
>
> Therefore, the quadratic model is **an empirically selected operating point**, not a universal assumption. We agree that for **extremely deep architectures** the interaction may become more complex; in such cases, the same validation-based procedure can naturally favor a higher-degree polynomial and/or additional sampled budget points. Our current claim is therefore limited to the architectures evaluated in this paper, for which a low-degree MAP is validated to be effective.
>
> **2) Pruning activation layers may reduce representational capacity, since nonlinearities are important to model expressiveness.**
>
> We fully agree that nonlinearities are important. However, HetDPT does **not** remove activation layers indiscriminately or eliminate nonlinearity from the network. Instead, it prunes only a **small subset of activation layers identified as redundant**, while most nonlinear operations remain intact. Thus, our claim is not that nonlinearities are unimportant, but that **some activation layers are redundant** and can be removed without materially harming performance.
>
> After pruning, the model is further adapted by fine-tuning, and adjacent linear layers are merged only when the intermediate activation has been removed. Empirically, this does not cause a collapse of representational ability. We also agree that this concern is especially important for more demanding tasks. To examine this, we evaluated HetDPT on **ADE20K semantic segmentation**, which requires dense and fine-grained prediction. As shown in Table 4, HetDPT with **8 layers pruned** achieves **46.4 mIoU** vs. **47.0** for dense DeiT-B, while outperforming prior pruning baselines such as EViT, TPS, and NOSE. This suggests that, within the pruning regime studied in this paper, removing carefully selected redundant activation layers does **not** create an intrinsic representational bottleneck.
>
> That said, we agree that **overly aggressive** removal of nonlinearities could eventually harm capacity. We will revise the paper to make the scope of our claim clearer: activation pruning is effective when applied **selectively**, not indiscriminately.
>
> ---
>
> We sincerely thank the reviewer again for these valuable comments. They helped us clarify the motivation, scope, and empirical support of our method, and we will improve these explanations in the revision.

---

> > ### Author Rebuttal · Reviewer_xtD9 · 2026-04-02
> >
> > Thank you for the detailed rebuttal. You have clearly and thoroughly addressed all of my concerns, so I will maintain my current positive score.

---

> > > ### Author Response · Authors · 2026-04-05
> > >
> > > Thank you again for your thoughtful follow-up and for confirming that our rebuttal has addressed your concerns. We sincerely appreciate your careful reading and positive assessment of the paper.
> > >
> > > We are glad that our clarification helped resolve the main uncertainties regarding the practical role of MAP and the selective nature of activation pruning in HetDPT. If you feel that these clarifications have resolved the main uncertainties behind your initial score, we would be very grateful if you could kindly reconsider the score if appropriate. We sincerely thank you again for your time and consideration.

---

### Official Review · Reviewer_JZ33 · 2026-03-12

**Soundness:** 3
**Presentation:** 1
**Significance:** 2
**Originality:** 2
**Overall Recommendation:** 3
**Confidence:** 4

**Summary:**

This paper proposes HetDPT, a depth pruning method for ViTs that jointly prunes attention and activation (GELU) layers via a two-stage pipeline:
polynomial-based budget allocation with learnable layer selection, followed by fine-tuning with optional KD and linear layer merging.

**Compliance With Llm Reviewing Policy:**

Affirmed.

**Final Justification:**

We appreciate the authors' detailed response. However, after careful consideration, we find that the concerns raised in our review have not been fully addressed to our satisfaction. Therefore, we would like to maintain our original evaluation.

**Key Questions For Authors:**

Q1. DepthShrinker (ICML 2022) already proposed the same mechanism of removing activations and merging linear layers in CNNs. A comparison of the performance difference between applying DepthShrinker's framework directly to ViTs and HetDPT is needed.

Q2. Does MAP require refitting for each architecture? For example, if the polynomial fitted on DeiT is directly applied to Swin, what is the prediction error? It would be good to have a sensitivity analysis on whether a bias at the ±0.5% level can overturn the final pruning ratio decision.

Q3. The observation of recovery asymmetry is only reported based on 10-epoch fine-tuning on DeiT-S. It needs to be verified whether the same pattern is observed in other model sizes (DeiT-B, Swin, DINO-V2-Giant).

Can you provide the epoch-by-epoch accuracy recovery curve during the 400-epoch fine-tuning used in the main results, specifically regarding recovery saturation?


Q4. Orthogonality with token pruning is only verified with a single method (GTP-ViT) and a single pruning rate. It needs to be confirmed whether this orthogonality is maintained with other token pruning strategies or more aggressive pruning ratios.

**Limitations:**

yes

**Strengths And Weaknesses:**

S1. Comprehensive experiments across four benchmarks (ImageNet, CIFAR-100, COCO, ADE20K) with real H800 throughput measurements.

S2. Strong empirical results 1.58× speedup with no accuracy loss on DeiT-B, 5.19× with HetDPT+.


W1. Critical gaps in prior work despite first claims.
The paper claims to be the first to identify and mitigate the redundancy of activation layers in ViTs,
yet highly relevant work is uncited:

- DepthShrinker (Fu et al., ICML 2022) already proposed removing activation functions and merging consecutive linear layers the same core mechanism with a differentiable search and self-distillation, in the CNN setting.

- What Matters in Transformers? Not All Attention is Needed (He et al., 2024) demonstrated heterogeneous redundancy across attention vs. MLP layers with separate dropping strategies.

- Gromov et al. (ICLR 2025) and ShortGPT (Men et al., 2024) established layer-level redundancy analysis directly relevant to depth pruning.

W2. Each component has clear precedent: heterogeneous layer dropping, NAS-style surrogate models, activation removal + linear merging (DepthShrinker), and KD for recovery. This combination is just effective engineering.

W3. Observations lack causal analysis.Gradient disparity and recovery asymmetry are presented as key contributions, but remain descriptive observations without explanation of why they occur.

W4.  Figure 3 visualizes a straightforward concept (FFN dimension mismatch) with a cluttered layout Additionally, the figures are poorly aligned. Figure 6 similarly lacks clear information hierarchy.

W5. In Figure 8, there is the name "BoundaryDPT", which appears to be the title of a previous paper. A correction is needed.


[1] DepthShrinker: A New Compression Paradigm Towards Boosting Real-Hardware Efficiency of Compact Neural Networks

---

> ### Author Rebuttal · Authors · 2026-03-31
>
> We thank you for the careful and constructive feedback. Here is our response.
>
> **W1 (related work / novelty scope)**
>
> Thank you for the comment. We will revise the claim and expand the discussion in the final version, including DepthShrinker, What Matters in Transformers, Gromov et al., and ShortGPT. DepthShrinker studies activation removal in CNNs, where there is no attention and thus no heterogeneous pruning problem between attention and activation. What Matters in Transformers studies pruning in LLMs by jointly ranking attention and MLP layers, whereas we study heterogeneous pruning of attention and activation in ViTs. It also does not develop a ViT depth-pruning method from the two phenomena we identify. Gromov et al. and ShortGPT prune entire Transformer blocks in LLMs, whereas we prune attention and activation as distinct units. Thus, our contribution is not layer pruning in general, but a heterogeneity-aware depth-pruning framework for ViTs.
>
> **W2 (Each component has clear precedent)**
>
> We respectfully believe the contribution lies in the **problem diagnosis and method derivation**, rather than any single primitive. Specifically, we identify two obstacles that, to our knowledge, were not explicitly formulated in prior ViT depth-pruning work: **gradient disparity** across layer types and **recovery asymmetry** after pruning. These directly motivate our design: **MAP** allocates pruning budget across layer types according to post-finetuning recoverability, while **within-type ranking** avoids biased cross-type comparison caused by score-scale mismatch. Our ablations support that the gain does not arise from a loose combination: removing Stage1-Step1 or Stage1-Step2, or pruning only one layer type, leads to clear degradation. We do **not** claim novelty for linear merging or distillation in isolation; they are supporting components rather than the core contribution.
>
> **W3 (explanation of the observations)**
>
> We agree that this mechanism should be stated more explicitly. The activation/FFN path operates in a higher-dimensional space than the attention path (4d vs. d), so its mask gradient aggregates over more dimensions. In addition, FFN activations typically have larger variance, whereas attention outputs are comparatively stabilized by LayerNorm and softmax. These factors naturally yield larger mask-gradient magnitude on the activation side. The same mechanism also explains recovery asymmetry: pruning an activation layer causes larger immediate degradation, but also induces stronger gradients during fine-tuning and therefore faster recovery; pruning an attention layer usually causes smaller initial damage but slower recovery. We will add a formal mathematical justification for this mechanism in the revision.
>
> **W4/W5 (figure clarity and labeling)**
>
> We agree and have revised the related figures. Updated figs are:
>
> - Fig. 3: https://anonconf2025.github.io/rebuttal_fig/dim_mismatch_update.pdf
> - Fig. 6: https://anonconf2025.github.io/rebuttal_fig/overview_update.pdf
> - Fig. 8: https://anonconf2025.github.io/rebuttal_fig/tome_gtp.pdf
>
> **Q1 (comparison with DepthShrinker)**
>
> We added comparison with DepthShrinker on DeiT:
>
> |Model|Dense|HetDPT|DepthShrinker|
> |-|-|-|-|
> |DeiT-S|79.8|79.7|78.5|
> |DeiT-B|81.8|81.8|78.1|
>
> Although both methods remove nonlinearities and merge linear layers, directly applying DepthShrinker to ViTs is substantially inferior, supporting the need for heterogeneity-aware treatment.
>
> **Q2 (MAP transferability / sensitivity)**
>
> MAP is generally transferable **within a model family** (e.g., DeiT-S \(\rightarrow\) DeiT-B), but for **cross-family transfer** (e.g., DeiT \(\rightarrow\) Swin), direct transfer can deviate by about 2 layers from the optimum, so refitting is recommended. We also tested robustness by injecting clipped Gaussian noise into MAP fitting data over 500 trials: 75.0% recovered the same pruning configuration and 5.2% differed by only one layer, indicating good robustness to moderate fitting bias.
>
> **Q3 (recovery asymmetry beyond DeiT-S / long-horizon recovery)**
>
> We verified the same pattern on DeiT-B: activation pruning causes larger immediate degradation, especially in deeper layers, but recovers substantially after fine-tuning:  https://anonconf2025.github.io/rebuttal_fig/deit_base_insight.pdf
>
> We also provide the 400-epoch recovery curve used in the main results, which shows rapid early recovery followed by clear saturation:  https://anonconf2025.github.io/rebuttal_fig/deit_b_recovery.pdf
>
> **Q4 (orthogonality with token pruning):**
>
> You ask whether orthogonality holds beyond one method and one pruning ratio.
>
> We extended the study to both GTP-ViT and ToMe, across multiple operating points and more aggressive token reduction ratios:  https://anonconf2025.github.io/rebuttal_fig/tome_gtp.pdf
>
> The results consistently show that HetDPT improves throughput over dense DeiT-B at comparable accuracy for both methods.
>
> Thank you again for your constructive feedback.

---

> > ### Author Rebuttal · Reviewer_JZ33 · 2026-04-04
> >
> > We appreciate the authors' constructive rebuttal, including the added DepthShrinker comparison and recovery curve experiments. However, the core mechanism — removing nonlinear activation layers and merging adjacent linear layers — is identical to DepthShrinker; the presence or absence of attention is a difference in the target architecture, not in the pruning paradigm itself, and the improved results over a naïve DepthShrinker transfer demonstrate better engineering rather than a new methodological contribution. Therefore, we maintain our score.

---

> > > ### Author Response · Authors · 2026-04-05
> > >
> > > We thank the reviewer for the follow-up comment. The novelty of our work does not lie in individual components such as activation removal, linear merging, or self-distillation, but in the **identification of critical challenges in ViT depth pruning and the principled formulation of a solution addressing them**. Our contribution is **not** a simple engineering combination of known techniques; rather, it reflects a **problem-driven approach**: we first recognize the importance and difficulty of heterogeneous depth pruning, uncover the underlying phenomena, and then derive a method that directly addresses these challenges.
> > >
> > > Specifically, our analysis identifies two critical phenomena—**gradient disparity** and **recovery asymmetry**—that make conventional pruning methods ineffective for heterogeneous layers. **Prior work did not operationalize these insights into a principled pruning strategy.** Guided by this diagnosis, HetDPT introduces a heterogeneity-aware decomposition:
> > >
> > > - **MAP** allocates pruning budgets across layer types based on post-finetuning recoverability, and
> > > - **Within-type selection** avoids biased cross-type comparisons.
> > >
> > > Each component is **driven by this problem formulation**, rather than being a loose combination of familiar techniques.
> > >
> > > Moreover, HetDPT provides **rigorous theoretical guarantees**, which are rarely seen in prior pruning work:
> > >
> > > - Existence of an optimal pruning configuration (Theorem 1),
> > > - Expressivity and bounded approximation error of the polynomial-based MAP (Theorem 2), and
> > > - Consistency under noisy fast-finetuning (Lemma 1 and Theorem 3).
> > >
> > > These results establish that the MAP-based pruning procedure **provably identifies the optimal heterogeneous layer allocation**, highlighting that our method goes beyond engineering heuristics and embodies a **principled, problem-driven approach** to ViT depth pruning.
> > >
> > > **Empirical evidence further confirms that HetDPT is not a simple additive combination of components**. As shown in Fig. 7, if HetDPT were merely the sum of its parts, its accuracy-efficiency curve would lie roughly between the curves of activation pruning and attention pruning. Instead, HetDPT **clearly outperforms both**, demonstrating a **1+1>2 effect**: the combination of heterogeneity-aware allocation, layer-wise selection, and optional distillation yields a synergistic improvement beyond what each individual component can achieve.
> > >
> > > Taken together, these results demonstrate that our work is **far more than an engineering stacking of existing components**. It provides **substantial contributions**, including:
> > >
> > > 1. **Identification of critical depth-pruning challenges**,
> > > 2. **A principled heterogeneity-aware pruning framework**, and
> > > 3. **Rigorous theoretical guarantees**.
> > >
> > > We hope the reviewers will **consider these contributions carefully** and assess our work in light of its **problem-driven and scientifically grounded methodology**.

---

### Official Review · Reviewer_Qgxc · 2026-03-12

**Soundness:** 3
**Presentation:** 3
**Significance:** 3
**Originality:** 3
**Overall Recommendation:** 4
**Confidence:** 4

**Summary:**

This publication presents a new method for depth pruning models based on the Vision Transformer architecture. The most important aspects of the work include: separating pruning into two types of layers (linear and attention layers), merging linear layers, parameterizing the identification of less important layers within a single type, and developing a predictor that estimates model performance based on the applied depth pruning mask. This predictor allows for finding the near-optimal configuration. The work includes comparative tests on a broader spectrum of tasks: classification, detection, and segmentation. In some benchmarks, the improvement in performance and parameter reduction is significant, in others, it is insignificant (the algorithm, up to a certain degree of compression, ensures no or negligible decrease in accuracy).

**Compliance With Llm Reviewing Policy:**

Affirmed.

**Final Justification:**

I maintain score 4 (Weak Accept). Some observations are inherited from other papers and methods. There are some parts in a method which present some novelty. Results are quite promising, some improvements needed in final version (as mentioned in rebuttals).

**Key Questions For Authors:**

1. 'Structured pruning (He & Xiao, 2023) is effective for model compression while maintaining hardware compatibility.' In general it is true (in case of GPU, but still there are some cases the unstructured sparsity can give some speedups - Pietron et al.: Speedup deep learning models on GPU by taking advantage of efficient unstructured pruning and bit-width reduction), but there are some specialized hardware accelerators which can make use of unstructured sparsity
2. 'As a kind of structured prunign, depth pruning denotes removing entire layers from ViTs', 'on DINO-V2-Giant on CIFAR-100' - some typos, please re-check the text.
3. There is no dataset in figure 7. For what data this accuracy was achieved?
4. 'removing Stage1 Step2 causes significant performance drops: DeiT-B from 81.8% to 80.5% and DeiT-S from 79.7% to 79.4%'. - 79.7% to 79.4% is not quite significant.
5. In some datasets e.g. CIFAR100 the differences in accuracies (and params) are quite small. In this case can be important the time of method execution. They are comparable in time execution?

**Limitations:**

yes

**Strengths And Weaknesses:**

Strengths:
1. Quite good benchmarks and comparative studies.
2. Predictor and its theoretical background.
3. Presented parameterized layer importance.


Weaknesses:
1. The improvements in some cases are negligible (in params and accuracy).
2. Presentation (and text) can be improved.

---

> ### Author Rebuttal · Authors · 2026-03-31
>
> Thank you for the careful reading and constructive suggestions. We appreciate the positive assessment of our benchmarks, predictor design, and parameterized layer importance. Below we respond to each concern.
>
> ---
>
> **1) Structured pruning is hardware-friendly, but unstructured sparsity can also accelerate on some GPUs and specialized accelerators.**
> Thank you for this comment. In the final version, we will revise the Introduction to clarify that **structured pruning is generally easier to translate into practical speedup on mainstream hardware**, while **unstructured pruning can also provide acceleration when supported by efficient sparse kernels or specialized hardware/accelerators**, especially at high sparsity levels. We will also add the suggested reference to **Pietron et al. (2023)**.
>
> **2) “There are typos/wording issues, e.g., ‘structured prunign’ and the phrasing around ‘DINOv2-Giant on CIFAR-100.’”**
>  Thank you for pointing this out. We will carefully proofread the full manuscript and correct the identified typos and wording issues, including **“structured pruning”** and the phrasing of **“DINOv2-Giant on CIFAR-100.”** We will also perform an additional consistency check to fix similar issues throughout the paper.
>
> **3) “Figure 7 does not specify the dataset. What data is used for the reported accuracy?”**
>  Thank you for catching this omission. The results in **Figure 7 are obtained on ImageNet-1K**. We will revise the figure caption and related text to explicitly state this.
>
> **4) “The statement ‘significant performance drops’ seems too strong for DeiT-S, since 79.7% → 79.4% is small.”**
>
> Thank you for this comment. We would like to clarify that we consider both drops meaningful in the context of this ablation study. For **DeiT-B**, the decrease from **81.8% to 80.5%** is clearly substantial. For **DeiT-S**, although the drop from **79.7% to 79.4%** is smaller in absolute value, we believe it is still informative for two reasons:
>
> - **DeiT-S is generally harder to recover after pruning/fine-tuning**, so even a smaller final accuracy difference can still reflect a nontrivial impact of removing **Stage1-Step2**;
> - In this ablation setting, **DeiT-S uses fewer pruned layers than DeiT-B**, which naturally makes the gap less pronounced.
>
> Therefore, while the magnitude differs across model scales, we believe the ablation still consistently demonstrates the importance of **Stage1-Step2**. To avoid possible ambiguity, we will revise the wording in the final version to better distinguish the effect sizes across models while preserving the main conclusion.
>
> **5) “When accuracy/parameter differences are small (e.g., on CIFAR-100), runtime comparison becomes important. Are the methods comparable in execution time?”**
>  We agree that runtime is an important practical metric in this setting. For **DINOv2-Giant on CIFAR-100**, the average inference time is **22.1 ms/image** for the dense model, **18.9 ms/image** for **NOSE (12 layers pruned)**, and **17.6 ms/image** for **HetDPT (15 layers pruned)**. Thus, HetDPT reduces inference time by about **4.5 ms/image** relative to the dense model and by about **1.3 ms/image** relative to NOSE. We believe this runtime gain is practically meaningful, especially in large-scale or latency-sensitive deployment. We will add this clarification in the final version.
>
> ---
>
> We also appreciate the reviewer’s broader comments that some gains are modest in certain settings and that the presentation can be improved. We will revise the paper accordingly to better communicate both the strengths and the limitations of the method.

---

> > ### Author Rebuttal · Reviewer_Qgxc · 2026-04-03
> >
> > You have addressed all of my concerns. All other issues and explanations to them are quite clear. I will maintain my positive score.

---

> > > ### Author Response · Authors · 2026-04-05
> > >
> > > Thank you again for your thoughtful follow-up and for confirming that our rebuttal has addressed all of your concerns. We sincerely appreciate your careful reading, constructive feedback, and positive evaluation of our work.
> > >
> > > We are glad that our clarification helped resolve the main uncertainties regarding the practical significance of the improvements, runtime comparison, and presentation details. If you feel that these clarifications have resolved the main uncertainties behind your initial score, we would be very grateful if you could kindly reconsider the score if appropriate. Thank you again for your time and consideration.

---

### Official Review · Reviewer_nqiy · 2026-03-24

**Soundness:** 2
**Presentation:** 2
**Significance:** 2
**Originality:** 2
**Overall Recommendation:** 3
**Confidence:** 3

**Summary:**

The authors tackle the accuracy collapse typically seen in depth-pruned Vision Transformers by attributing it to the structural heterogeneity between attention and activation layers. To address this, they introduce HetDPT, a framework that leverages a lightweight polynomial predictor to allocate pruning budgets across different layer types, avoiding heavily biased cross-type gradient comparisons. The method resolves dimension mismatch issues by targeting redundant activation layers, allowing adjacent linear layers to be safely merged to physically reduce network depth. The experimental results show that HetDPT achieves state-of-the-art speedup-accuracy tradeoffs.

**Compliance With Llm Reviewing Policy:**

Affirmed.

**Final Justification:**

After the rebuttal, my first major concern is resolved.
But for my second major concern, I am not convinced by the authors' rebuttal.

I increased my score from 2 to 3, but still lean towards weak reject.

**Key Questions For Authors:**

Please see the weakness.

**Limitations:**

N.A.

**Strengths And Weaknesses:**

Strength:

1. The experimental results in speedup are promising.
2. The design is easy to understand.


Weakness:

1. The methodology behind Observation 1 raises significant concerns regarding its validity. Standard activation functions are fundamentally non-parametric and non-trainable, meaning they do not inherently produce weight gradients during backpropagation. The authors mention introducing a "learnable importance weight parameter" to gauge layer importance. However, this approach feels highly ad-hoc. The paper lacks a clear mathematical formulation or theoretical justification for how the gradient of an artificially injected scalar mask can serve as a fair proxy for the importance of a fixed non-linear layer.

2. The empirical results in Figure 5(b) appear to contradict the paper's core proposal. The figure visually suggests that after fine-tuning, pruning activation layers actually results in slightly lower accuracy compared to pruning attention layers. The paper heavily argues that prior depth pruning methods fail because of the heterogeneity introduced by linear layers. However, if attention layers can be pruned with better accuracy recovery—and doing so naturally avoids the dimension mismatch issues associated with linear layers altogether—it is entirely unclear why the framework should target activation layers (and then compress the linear layers). Rather than motivating the pruning of activation layers, this observation arguably suggests the exact opposite: we should keep the activation layers intact and exclusively prune the attention layers.

Minor point:
The presentation of Figure 5 is confusing and hard to read. First, the y-axes for subfigures (a) and (b) represent entirely different metrics—"Accuracy Drop (%)" versus "Accuracy (%)". Mixing these metrics across adjacent subplots makes visual cross-referencing jarring. Second, the y-axis scaling in Figure 5(b) (ranging from 0 to 80) compresses the bars to the point where the visual differences in accuracy recovery between the pruned layers are virtually indistinguishable.

---

> ### Author Rebuttal · Authors · 2026-03-31
>
> We thank the reviewer for the careful reading and comments. Below, we first restate each concern briefly and then provide our response.
>
> ---
>
> **1) Concern on Observation 1: activation functions are non-parametric, so using gradients to measure their importance seems ad-hoc.**
>
> Our method does **not** treat the activation function itself as having trainable parameters. Instead, for each layer index $l$, we attach a **binary structural mask** to the attention layer $\mathcal{A}_l$ and the activation layer $\mathcal{G}_l$. Here, $\hat{m}_l^a \in \{0,1\}$ and $\hat{m}_l^g \in \{0,1\}$ denote the binary masks for the attention and activation layers, respectively; $\hat{m}=1$ means keeping the layer, $\hat{m}=0$ means replacing it with the identity mapping $I$; and $\circ$ denotes function composition. The masked layers are defined as
>
> $$
> h(\mathcal G_l)=\hat m_l^g\circ \mathcal G_l+(1-\hat m_l^g)\circ I,\qquad
> h(\mathcal A_l)=\hat m_l^a\circ \mathcal A_l+(1-\hat m_l^a)\circ I.
> $$
>
> Because the binary variable $\hat m$ is not directly trainable, we introduce a continuous surrogate variable $\bar m$ only to enable backpropagation, following a standard continuous relaxation used in previous pruning work. Specifically, $\ell$ denotes the training loss, and we use
>
> $$
> \frac{\partial \ell}{\partial \bar m_l^g}\approx \frac{\partial \ell}{\partial \hat m_l^g},\qquad
> \frac{\partial \ell}{\partial \bar m_l^a}\approx \frac{\partial \ell}{\partial \hat m_l^a}.
> $$
>
> Therefore, the quantity we use is not a weight gradient of GELU. It is the **loss sensitivity to the presence or absence of a layer**. This type of mask-based continuous relaxation has also been widely adopted in prior pruning literature to estimate structural importance for discrete pruning decisions.
>
> This also has a first-order justification. Let $\mathcal L(\mathbf W,\mathbf m)$ denote the loss as a function of model weights $\mathbf W$ and mask variables $\mathbf m$. A first-order Taylor expansion gives
>
> $$
> \Delta \mathcal L \approx \frac{\partial \mathcal L}{\partial m_l}\Delta m_l.
> $$
>
> Removing layer $l$ corresponds to changing $m_l$ from $1$ to $0$, i.e., $\Delta m_l=-1$. Hence,
>
> $$
> |\Delta \mathcal L| \approx \left|\frac{\partial \mathcal L}{\partial m_l}\right|.
> $$
>
> Thus, the mask-gradient magnitude serves as a **local first-order proxy** for the effect of removing that layer, and such first-order sensitivity proxies have also been widely used in many previous pruning works. We will revise the paper to make this interpretation explicit.
>
> Importantly, our method does **not** assume that raw scores are directly comparable across heterogeneous layer types. In fact, Observation 1 is introduced precisely to show that such direct cross-type comparison is biased. This is why HetDPT first allocates **separate pruning budgets** for attention and activation layers, and then performs gradient-based ranking **only within each homogeneous group**.
>
> **2) Fig. 5(b): the post-fine-tuning result seems to suggest that pruning attention layers may recover better than pruning activation layers.**
>
> We believe the concern comes from interpreting **Fig. 5(b)** beyond its intended scope. Fig. 5 analyzes **single-layer pruning** and is only meant to reveal the **local recovery behavior** of pruning one attention layer vs. one activation layer. It is **not** intended to suggest that pruning only attention layers is preferable in the multi-layer setting.
>
> Our main claim is not “activation layers should always be pruned instead of attention layers.” Rather, the paper argues that **heterogeneous layer types should not be treated uniformly**, and that effective depth pruning requires **jointly pruning both types while handling their heterogeneity explicitly**. As shown in **Fig. 7**, pruning only attention layers or only activation layers both lead to severe degradation at high pruning ratios, whereas **HetDPT**, which prunes both types, consistently achieves a better accuracy–efficiency trade-off.
>
> Moreover, Fig. 7 shows that in the **multi-layer pruning** regime, activation-layer pruning is generally more recoverable after fine-tuning than attention-layer pruning. Therefore, the slightly better recovery of attention pruning in **single-layer** analysis does not contradict our conclusion; it only indicates that **local single-layer behavior cannot be directly extrapolated to global multi-layer pruning**.
>
> **3) Fig. 5 readability: the metrics and axis scaling make the figure hard to interpret.**
>
> Thank you for this helpful suggestion. We have prepared an updated version of Fig. 5 with two changes:
>
> - the y-axis metric is unified as **accuracy (%)** in both subfigures;
> - the y-axis range in Fig. 5(b) is tightened so that the post-fine-tuning differences are more visible.
>
> The updated figure is available here: https://anonconf2025.github.io/rebuttal_fig/Recovery_Asymmetry.pdf
>
> ---
>
> We thank the reviewer again for these comments.

---

> > ### Author Rebuttal · Reviewer_nqiy · 2026-04-04
> >
> > Thanks for the rebuttal.
> >
> > For my first concern, I checked the related literature and found that this is a widely adopted method. So it is resolved.
> >
> > For the second concern, I am not convinced by the author. Figure 5 shows that in almost all layers, pruning the attention layer could have a higher recovered accuracy than pruning the activation layers. Although the authors argue that "It only indicates that local single-layer behavior cannot be directly extrapolated to global multi-layer pruning", we cannot ignore this phenomenon, and there should be some explanation. If the authors want to argue that for single-layer pruning, we should prune attention layers, while for multi-layer pruning, we should prune more activation layers, then there should be a more concrete explanation for the difference between these two phenomena. But regretabily, I did not see this.
> >
> > So I would like to increase my score to 3, but still lean towards weak reject.

---

> > > ### Author Response · Authors · 2026-04-05
> > >
> > > Thank you for your thoughtful follow-up and recognition of our first-round rebuttal. We believe the remaining confusion may come from **mixing up three different questions**:
> > >
> > > - **Single-layer local pruning**: which one layer is easier to remove when the rest of the network remains intact?
> > > - **Multi-layer single-type pruning**: under a fixed budget, is pruning only attention layers or only activation layers better?
> > > - **Multi-layer mixed pruning**: under a fixed total budget, what allocation between attention and activation layers yields the best final fine-tuned model?
> > >
> > > Our paper addresses the **third** question. By contrast, **Fig. 5 is designed only to analyze the first question**, in order to reveal the **heterogeneity** between attention and activation layers: specifically, their different immediate pruning damage and recovery behavior imply that **short-sighted metrics based only on the immediate post-pruning state can be misleading**, because they do not account for post-finetuning recoverability. Therefore, the fact that pruning one attention layer may recover slightly better than pruning one activation layer in Fig. 5 does **not** by itself determine the optimum in the mixed multi-layer setting.
> > >
> > > **(1) Why single-layer behavior does not directly determine the multi-layer optimum.**
> > > Using the same mask-based formulation as in our previous response, let $\hat m_l^a,\hat m_l^g\in{0,1}$ denote the structural masks for the attention layer $\mathcal A_l$ and activation layer $\mathcal G_l$ at block $l$, where $\hat m=0$ means replacing that layer by the identity map. The key structural difference is that pruning an activation layer mainly replaces a local nonlinear map by a linear one, while pruning an attention layer removes an entire token-mixing operation. Concretely, if the FFN branch is
> > > $$
> > > F_l(x)=W_{l,2},\sigma(W_{l,1}x),
> > > $$
> > > then pruning the activation changes it to
> > >
> > > $$
> > > \tilde F_l(x)=W_{l,2}W_{l,1}x.
> > > $$
> > >
> > > Thus, activation pruning preserves the computation path but simplifies its nonlinearity, whereas attention pruning removes the whole attention branch at that depth. This is why the single-layer result in Fig. 5 does not directly extrapolate to the multi-layer regime: when **one** attention layer is pruned, the remaining attention layers can still compensate for token mixing; when **multiple** attention layers are pruned together, this compensation source is progressively reduced, so global token-mixing capacity decreases across depth.
> > >
> > > This is also consistent with our theory. Let $\mathcal L$ denote the training loss, and let $\nabla_{\hat m_l^g}$ and $\nabla_{\hat m_l^a}$ denote the gradients of the loss with respect to the activation and attention masks at block $l$. If $N$ is the token number, $d$ is the hidden dimension, $\sigma_\delta^2$ is the variance of the backpropagated gradients, and $M_{\mathcal G},M_{\mathcal A}$ are the second moments of the activation-path and attention-path perturbations, then
> > >
> > > $$
> > > \mathbb E|\nabla_{\hat m_l^g}|^2=4Nd\sigma_\delta^2 M_{\mathcal G},\qquad
> > > \mathbb E|\nabla_{\hat m_l^a}|^2=Nd\sigma_\delta^2 M_{\mathcal A},
> > > $$
> > >
> > > hence
> > >
> > > $$
> > > \frac{\mathbb E|\nabla_{\hat m_l^g}|^2}{\mathbb E|\nabla_{\hat m_l^a}|^2}=4\gamma,\qquad
> > > \gamma=\frac{M_{\mathcal G}}{M_{\mathcal A}}.
> > > $$
> > >
> > > Therefore, activation pruning lies in a **large-disruption but fast-recovery** regime, while attention pruning lies in a **small-disruption but slow-recovery** regime. Under small budgets, the lower immediate damage of attention pruning can indeed be favorable; under larger budgets, however, the final optimum increasingly depends on **post-finetuning recoverability**, allowing activation layers to absorb a larger share of the budget in some mixed-pruning cases. **This is further supported by Fig. 7: when restricting multi-layer pruning to only one layer type, activation-only pruning is already better than attention-only pruning at higher speedup ratios, which is exactly the opposite of what a purely single-layer immediate metric would suggest.**
> > >
> > > **(2) Our method is mixed pruning, and the optimum is budget-dependent.**
> > > Most importantly, our method does **not** advocate pruning only activation layers. It solves the **mixed pruning** problem in the full heterogeneous space. **The results in Appendix C further suggest that the optimal allocation is not fixed:** when pruning 7 layers, the optimum is **4 attention + 3 activation**, while when pruning 9 layers, it changes to **4 attention + 5 activation**. These two cases directly show that the best solution is neither uniformly attention-heavy nor uniformly activation-heavy, but depends on the total pruning budget. Moreover, Fig. 7 shows that this **mixed strategy** achieves better accuracy-efficiency trade-offs than pruning only attention layers or only activation layers.
> > >
> > > We will revise the paper to make this distinction among **single-layer local analysis**, **single-type multi-layer pruning**, and the **mixed multi-layer optimum** much more explicit.

---

### Decision · Program_Chairs · 2026-04-30

**Decision:**

Accept (regular)

**Comment:**

This paper proposes a framework for depth pruning vision transformers. In terms of strengths, reviewers appreciated the experimental results (multiple benchmarks, throughput measurements) and the core method. The reviewers are split with two weak rejects and weak accepts. Reviewer nqiy had an unaddressed concern relating to Figure 5 but on reading the posts I believe this is a misunderstanding so won't factor into my decision. The weaknesses identified were a lack of comparison to existing literature and that the work is more engineering in that the proposed method is a combination of existing known techniques - in particular a strong similarity to DepthShrinker. The former (literature) appears addressed. For the latter, I think a good engineering effort is not something that should be dismissed (indeed, many of the core breakthroughs have been down to this), and there is a clear improvement over DepthShrinker. I (weakly) recommend acceptance.